# On the seasonal variation of observed size distributions in Northern Europe and their changes with decreasing anthropogenic emissions in Europe: climatology and trend analysis based on 17 years data from Aspvreten, Sweden

Peter Tunved and Johan Ström

Department of Environmental Science and Analytical Chemistry, Stockholm University, 10691 Stockholm, Sweden

Correspondence to: Peter Tunved (peter.tunved@aces.su.se)

**Abstract.** Size resolved aerosol trends were investigated based on a 17-year data set (2000-2017) from the rural background
site Aspvreten located in southern Sweden (58.8°N, 17.4°E). Cluster analysis of the size distributions was performed to aid in the interpretation of the data. The results confirm previous findings of decreasing aerosol mass and number during last decades as a result of reduced anthropogenic emissions in Europe. We show that both particle modal number concentration and size substantially has been reduced during last 17 years. Negative trends in particle number concentration of about 10 $cm^{-3} y^{-1}$ is present for nuclei, Aitken and accumulation modes. In total, integral particle number concentration has decreased
by 30 %, from 1860 $cm^{-3}$ to ca 1300 $cm^{-3}$. The reduction in modal number concentration is accompanied by a decrease in modal size, and this decrease is largest for the accumulation mode (2 nm $y^{-1}$ or about 17 % for the whole period). These reductions have resulted in a decrease in submicron particle mass (<390 nm) by more than 50 % over the period 2000-2017. These decreases are similar to observations found at other stations in Northern Europe.

Although all size classes show a downward trend as annual averages, we also show that observed trends are not evenly
distributed over the year, and that a rather complex picture emerges where both sign and magnitude of trends vary with season and size. The strongest negative trends are present during spring (accumulation mode) and autumn (Aitken mode). The strongest positive trends are present during summer months (Aitken mode). The combined trajectory and data analyses do not present evidence for an increase in new particle formation formed locally, although some evidence of increased new particle formation some distance away from the receptor is present. Observed aerosol size distribution data, together with an
adiabatic cloud parcel model, was further used to estimate the change in cloud droplet concentration for various assumptions of updraft velocities and aerosol chemical composition. The results indicate a substantial increase in the atmospheric brightening effect due to a reduction in cloud reflectivity corresponding to 10-12 % reduction in cloud albedo over the period 2000-2017.

# 1.    Introduction

Turbidity of the atmosphere is a result of light being scattered and absorbed by particles suspended in the air. For most ambient conditions there is a strong proportionality between the light scattered and the total particle volume in the accumulation mode range between about 0.1 and 1 μm diameter (Willeke and Brockmann, 1977). This size dependence arrives from the fact that the wavelength of the peak in solar radiation is in the same size range as the accumulation mode particles. Hence, variations in the amount of accumulation mode particles are directly affecting the turbidity of the atmosphere.

An increase in aerosol loading tends to scatter more of the incoming solar radiation back to space, the so called "dimming effect", which leads to less radiation reaching the Earth's surface and a net cooling of climate system (Myhre et al., 2013). Observations of solar radiation incident at Earth's surface indicate that global dimming increased up to about 1990, but after that started to decrease, the so called "brightening effect" (Wild et al., 2006). Streets et al. (2006) suggested that this trend in global dimming/brightening was due to the combined effect of economic growth and the recent decrease in emission of aerosol particles and their gaseous precursors as a result of legislative measures.

The most important aerosol precursor is sulfur dioxide, which has both natural and anthropogenic sources. Sulfur dioxide is oxidized to sulfuric acid vapour and later transforms into sulfate aerosol mass, fully or partially neutralized by ammonia. Sulfuric acid vapour participates both in new particle formation and contributes to aerosol mass concentration by condensation on already existing particles. Abovementioned air quality policies mainly targeted sulfur emissions which have led to a general decrease in the anthropogenic emissions of $SO_2$ since the 1970's (Smith et al., 2011;Klimont et al., 2013). With perhaps the exception of the Indian sub-continent, this decrease appears to continue also through the last decade (Li et al., 2017; US EPA, 2018).

Zhao et al. (2017), showed that the decrease in $SO_2$ and other important anthropogenic emissions, such as nitrogen oxides and carbonaceous material, is consistent with the decrease in the trends in dimming observed from space borne instruments. They studied three regions, eastern United States, Western Europe, and Eastern and Central China, and concluded that the trends in aerosol optical depth (AOD) are consistent with the trends in emissions for these three regions.

Coen et al. (2013), studied trends in observed aerosol optical properties measured at ground-based stations mainly located in North America and Europe. They concluded that, even if the trends are not homogeneously distributed geographically, the decreasing trend observed for most stations in North America is related to the decrease in anthropogenic emissions of particles and their gaseous precursors. The pattern of trends over Europe was not as clear as over North America and the Arctic and Antarctic stations did not show significant trends at all.

On the other hand, Turnock et al. (2015) found that the reduced emissions have led to a substantial decrease in the concentration of aerosols over Europe. They used a global environmental model to consolidate the observed trends in aerosol loading over Europe and changes in emissions. They also explored the trends in the microphysical properties of the aerosol by comparing the model to observed number densities of different size ranges. They concluded that the model under-predicts

the observations in general, and in particularly this was the case for the two Arctic stations included in the study. The authors did not look at the trends for different particle size ranges per se but focused on how the model reproduced the data at the various sites.

By studying long-term observations of aerosol microphysical data available through the GAW Global Atmospheric Watch (GAW) program, organized by the World Metrological Organization (WMO) and the European Union infrastructure project ACTRIS, Asmi et al. (2013) explicitly studied the decadal trends in total aerosol number concentrations. They came to a similar conclusion as the studies above, that the most likely cause for the generally decreasing trends in aerosol number concentrations in the Northern hemisphere is the decrease in anthropogenic emissions of $SO_2$ or other co-emitted species. To address the question about possible trend in cloud condensation nuclei (CCN) particles, the authors divided the observed size distribution into to number densities of particles larger than 20 nm (N20) and 100 nm (N100), respectively. Here, N100 was a proxy for CCN particles. However, due to claimed lack of stations with more than 10 years of data, only five stations were available for their study. From this analysis they made two interesting observations. The first observation was that the trends in both N20 and N100 were consistent at the four sites (three stations in Finland and one in Sweden), which led the authors to conclude that the trends are uniform over the size distribution. The second observation was the lack of a diurnal difference in trends. This led the authors to hypothesize that either the characteristics of the size distributions are mainly controlled by long-range transport, or the frequency or intensity of new particle formation has stayed rather constant over the investigated time period.

The statistics of the occurrence of nucleation events over an eight-year period was explicitly studied by Dal Maso et al. (2005). This study was conducted for one of the Finnish stations participating in the study by Asmi et al. (2013), namely the Hyytiälä SMER II station. The authors do not explore any possible trends in the occurrence of new particle formation, but their Figure 6 suggests that between 1996 and 2003 the number of event-days (days with clear new particle formation) and undefined (cases with small particles but not a clear event-day) increases over time, while clear non-event days suggests little change. How this potentially small increase in events is distributed over the year is not possible to assess from their study.

Clearly there is a lack of long-term aerosol microphysical data from which trend analysis has been reported or even can be reported because of too short measurement periods or interrupted data sets. Trend analysis require long term uninterrupted time series to be meaningful. This does not mean that data is completely missing. Through collaborations within European Supersites for Atmospheric Aerosol research (www.EUSAAR.net), long term standardized observations of e.g. aerosol number size distribution observations are available from several European sites, some of which are long enough to support time series analysis and determination of trends (e.g. Asmi et al., 2013). The characteristics of the general properties of aerosols are better known as there exist a large quantity of studies addressing the different aspects of the aerosol number size distribution observed in Fenno-Scandia. In this region, several long term monitoring stations exist covering vegetation zones from nemoral to polar sites north of the tree-line, e.g. the southern Sweden nemoral site Vavihill (56.01°N, 13.09°, 172 m asl) (Tunved et al., 2003;Kristensson et al., 2008), the southern boreal site Hyyiälä (61.85°N, 24.29°, 179 m asl) (Dal Maso

et al., 2005; Kulmala et al., 2008) and the northerly located stations Värriö (67.76°N, 29.61°, 390 m asl) and Pallas (67.97°N, 24.12°, 560 m asl) (Laakso et al., 2003;Tunved et al., 2003;Dal Maso et al., 2007). Available reports show that both the shape and number concentration over Scandinavia are highly variable. At northerly located stations such as Värriö and Pallas (~67°N), reported annual average number concentration of nuclei mode particles are around 50-100 cm$^{-3}$, Aitken

mode particles ca 200 cm$^{-3}$ and accumulation mode particle concentration ~150 cm$^{-3}$ (Tunved et al., 2003;Komppula et al., 2006;Dal Maso et al., 2007). Observations at Hyytiälä (61°51'N) yield averages of ca 300 cm$^{-3}$ nucleation mode size particles, and 750 cm$^{-3}$ and 290 cm$^{-3}$ in Aitken and accumulation mode size range, respectively. In southern Sweden, (Kristensson et al., 2008) reports accumulation mode concentration around 300 cm$^{-3}$ and Aitken mode concentration around 800 cm$^{-3}$. At all stations above, there is a strong seasonality in aerosol size distribution properties, with a typically bimodal

wintertime size distribution with on average low number concentration. This is contrasted by a near unimodal summertime size distribution with high total number concentration. It has further been demonstrated in studies of air-mass transport across Scandinavia, that the Boreal region act as a net source of particle number (and mass) during southerly transport of marine clean air (Tunved et al., 2006a;Tunved et al., 2006b;Vaananen et al., 2013), while acting as a relative sink when polluted air is transported northwards. During the southerly transport, it is believed that the forest is playing an active role in

particle formation and growth, via nucleation and condensation. During northerly transport, transport of air-masses with a comparable high aerosol condensation sink leads to rarer nucleation events and sink processes, especially wet deposition, leads to reduction in number and mass.

How these mean aerosol characteristics change over time may contain surprises, as pointed out by Julin et al. (2018). Using a chemical transport model, the authors showed that aerosol properties in the Nordic countries may have a very different

temporal evolution compared to central Europe with respect to the total number density of particles. The model simulations suggest that changes in aerosol properties may not only follow the reduction in emissions, but the number density of particles may even increase as a result of legislative actions. The explanation for this is an already relatively low emission of precursor gases together with a general decrease in the condensational and coagulation sink by pre-existing particles in neighbouring countries, which sets the stage for new particle formation (Kulmala et al., 2001; Kerminen and Kulmala,

25   2002).

Based on the literature referenced above, several locations exhibit a decrease of aerosol mass observed overpopulated areas in the last decades and this decrease is related to the decrease in emissions of anthropogenic primary particles and precursor gases. There is also a suggestion that this decrease may not be uniformly distributed over the whole size distribution. In our opinion it is important to establish how this reduction is distributed in order to understand the implications of the reduction in

emissions, which means that the whole size distribution must be studied. There might be a general decrease in aerosol loading that is shared by all particle sizes, but if one type of aerosol size distribution becomes less frequent, this must be compensated by some other type of size distribution (e.g. accumulation mode, Aitken mode, or nucleation mode dominated size distributions).

We have chosen to study hourly averaged size distributions observed at Aspvreten, Sweden. The data extends over a 17-year period from June 2000 through November 2017. The measurement site at Aspvreten is characterized as a rural background station. Our aim is to study the aerosol size resolved trends and try to establish how the characteristics of the aerosol size distributions have changed over time as a potential result of reduction in anthropogenic emissions. We will explore seasonal differences and possible changes in transport patterns. To test how potential changes in aerosol properties at Aspvreten may influence cloud droplet number concentrations, a simple parcel model was used to simulate cloud droplet activation. The model was initiated using observed size distributions for a range of updraft velocities and characteristic chemical aerosol properties.

## 2.     Methods

### 2.1     Aerosol number size distribution

Aspvreten observation station (58.8°N, 17.4°E, 25 meters above sea level) is located in the county of Södermanland, about 80 km south of Stockholm. The station is situated close to the Baltic Sea, and the coastline is located a few km to the East and about one km to the south of the station. The surroundings are dominated by deciduous and conifer forests, in mosaic with farmlands. The station represents typical continental rural background conditions with few local pollution sources or densely trafficked roads. Climatologically, the surroundings represent the boreal-nemoral zone, a transition region between the temperate southern nemoral zone (mainly deciduous broadleaf forest) and the boreal zone.

This study focuses on aerosol number size distribution observations collected at Aspvreten observation station using a differential mobility particle sizer (DMPS). The station has been housing aerosol number size distributions since year 2000 and Stockholm University have performed observations of aerosol number size distribution properties with near identical instrumental setup for almost two decades. In 2018 the measurements were moved to Norunda, which is about 90 km north of Stockholm and these data are not included in current analyses. Nevertheless, the DMPS data set from Aspvreten is one of the longest in Europe. The instrument setup consists of a medium differential mobility analyzer (DMA) with a TSI 3010 condensation particle counter (CPC), and the system is capable of monitoring the aerosol number size distribution between 10 and roughly 400 nm with a temporal resolution of around 15 minutes. The actual length of each scan varied slightly over the years, but our base data will be hourly averages, which makes this a minor issue.

More details on the instrumental setup and station characteristics may be found in (Tunved et al., 2003;Tunved et al., 2004; Tunved et al., 2005)  The station, being a part of the European Supersites for Atmospheric Aerosol Research project (EUSAAR http://www.eusaar.net/), has undergone audits to assure the application of harmonized measurement and sampling protocols regarding aerosol number size distribution.

In this study, we utilize all the data recorded at Aspvreten, from 2000 to 2017. After screening for instrument failures and maintenance periods, the data coverage is typically better than 75 % on annual basis. The only years when less than 75 %

data coverage occurred was for the starting year 2000 (the year of the measurement start-up, with 25 % data coverage), for 2002 (70 % data coverage) and for 2010 (50 % data coverage). In this study, we have made use of hourly average size distributions, and a total of 132182 hourly data points have been calculated, equivalent of more than 5500 days of data, or 83 % coverage for the whole period. Since the actual bin sizes of the instrument have changed slightly over the years, all data

have been harmonized to a fixed logarithmic equidistant size grid distributed over 34 bins between 10 and 390 nm. Previous studies using aerosol number size distribution observations from Aspvreten include (Tunved et al., 2003;Tunved et al., 2004;Tunved et al., 2005;Dal Maso et al., 2007;Dal Maso et al., 2008). These studies have on mainly focused new particle formation, seasonality, lifecycle and transport characteristics of the Fennoscandinavian aerosol. The current study is the first decadal trend analysis based on aerosol size distribution data from Aspvreten.

The availability of a long time series (17 years), which have used near identical instrument set-up during the full operational time, makes this data set most suitable for the study of long term trends of aerosol size distribution properties, on par with other long term archives such as the approximately 20 year aerosol size distribution data set from Hyytiälä (e.g. Asmi et al., 2013).

During the entire period of interest, the instrument has been maintained by the same technician. This by itself assures

consistent calibration routines and standard operating procedures. Further, the DMPS-system have been part of the European Supersites for Atmospheric Aerosol Research (EUSAAR, http://www.eusaar.net/) intercalibration workshop and the setup follows the recommendations made by EUSAAR regarding sampling and inversion. This certifies that any error introduced in the trend analysis by changes in sampling routines quality assurance are minimized.

## 2.2    Log-normal fitting

All hourly aerosol size distributions were fitted assuming that the aerosol can be described by the sum of log-normally distributed modes. We decided a priori to fit the distribution over three modes after concluding that three modes in most cases were sufficient to capture the size distribution properties over the available size range. Equation (1) gives the general

form of a log-normal mode,

$$n_N \left( \log D_p \right) = \frac{N}{(2\pi)^{1/2}} \exp\left( -\frac{(\log D_p - \log \overline{D}_{pg})^2}{2 \log^2 \sigma_g} \right), \qquad\qquad (1)$$

where $n_N$ is the number size distribution function, $N$ is the number concentration of the mode, $D_p$ is the size of the particle

diameter, $\overline{D}_{pg}$ is the geometrical mean diameter of the mode and $\sigma_g$ is the geometric standard deviation, henceforth denoted GSD, giving an estimate of the width of each log normal mode centred around $\overline{D}_{pg}$.

The fitting was performed utilizing the Matlab function fmincon.m to perform a constrained fit of the three modes over the size range covered by the instrument. The number concentration of each mode is allowed to vary from 0 to infinity, and the GSD is allowed to vary between 1.1 and 2.5. When fitting, we do not force the modes into predefined size ranges, i.e. typical nuclei, Aitken or accumulation mode size ranges, but instead allows the algorithm to find the best numerical solution over the whole size range covered by the instrument. After fitting has been performed, the modal parameters are arranged according to size ($\overline{D}_{pg}$), resulting in modes 1-3, with mode number one being the smallest and mode number three being the largest. The modal sizes derived in this way roughly corresponds to nuclei, Aitken and accumulation modes.

## 2.3    Cluster analysis

Clustering of aerosol size distribution have proved to be a very useful tool in studies of aerosol lifecycle and different process studies, as demonstrated in several studies following Tunved et al. (2004), including e.g. (Beddows et al., 2009; Wegner et al., 2012;Freud et al., 2017; Rizzo et al., 2018) in rural, urban, and Arctic environments.

Contrary to standard averaging of number size distributions, cluster analysis and associated centroids can conserve the shape of the aerosol size distribution. Hence, size distribution clusters represent "signature distributions" that reflect contribution from members that are likely to have undergone similar processing in the atmosphere prior to observations. Thus, clustering size distribution and combining the cluster analysis with auxiliary parameters, such as trajectory derived source areas, temporal distribution of members and parameters related to sink processes (e.g. precipitation) can provide a deeper insight into the multitude of factors defining the aerosol over time.

In this study we have applied a Matlab version (Statistics and machine learning toolbox) of kmeans clustering (kmeans.m) to perform a clustering of the hourly size distributions. Naturally, there are several ways to determine the optimum number of clusters to use in order to maximize the inter cluster variability while at the same time minimizing the intra cluster variability. Various functions are available in Matlab for this, e.g. the silhouette.m function. Although similar cluster evaluation algorithms rely on sound mathematical foundations, in the end a subjective decision must be done in order to decide the criteria used, and this criterion should reflect the needs with respect to level of detail of the project at hand. In Tunved et al. (2004), eight clusters were used to characterize one year of data from Aspvreten. For the trend analysis in this study, we increased the number of clusters to 12, in order to better capture changes between clusters over time of the much larger data set. Test were performed with both more and fewer clusters. The best balance between information content and data amount to be presented was found to be reasonable around twelve clusters. A balance of information content refers to be able to follow a logical context between different clusters, but also that the whole set of clusters can characterize the domain of different size distributions observed at Aspvreten.

The clustering was performed on hourly averaged data, using options "max iterations" of 10000 and "number of replicates" was set to 10 in Matlab. The distance function applied was squared Euclidean distance, assuming that the difference is calculated from the centroids defined as the mean of the points in the clusters,

$$d(x,c) = (x - c)(x - c)',$$ (2)

where x is an observation (i.e. the size distribution vector) and c is the centroid.

## 2.4 Trajectory analysis

In order to explore source dependence of aerosol properties, and to allow to study the effect different meteorological parameters have on the observed size distribution characteristics, air mass back trajectories were calculated using the Hysplit4 model (Draxier and Hess, 1998). Trajectories were calculated on hourly basis, with a total trajectory length of 240
hours. Arriving altitude was set to 100 m above ground level, and model derived relative humidity, precipitation amount, mixing layer height and temperature were saved for each trajectory point. The meteorological input used by the model is the FNL archives for the time period 2000-2005, and the GDAS1deg for the time period 2006-2017. The main difference between these two data sets lies in the resolution; GDAS1deg is given with a one-degree resolution, while the FNL dataset uses a resolution of about 191km. The FNL archive is no longer updated and was replaced by the one-degree GDAS dataset
in 2005. Another difference is that the FNL data is provided as hemispheric data, two files per month, while the GDAS1deg is provided as weekly files with global coverage (for details cf. https://www.ready.noaa.gov/archives.php).

## 3. Results

## 3.1 General seasonal size distribution properties

As discussed above, the aerosol over the Fennoscandinavian region is highly variable, and observations show a strong seasonality in aerosol size distribution properties. Figure 1 presents the average seasonal variation of all validated number size distributions for the period 2000-2017. Integral number is superimposed on the surface plot as median and 25th-75th percentile range. As can be seen, a strong seasonality is present. The lowest number concentration is found in November and December whilst the highest number concentration is found during summer. It is also evident that the changes of aerosol
number size distribution occur gradually and the transition between the seasons are smooth, quite opposite to the seasonal characteristics of for instance the Arctic site Zeppelin (Tunved et al., 2013). The seasonally averaged hourly number size distributions are presented in Figure 2 as complement to Figure 1. Figure 2 shows seasonal median number size distribution for the whole period 2000-2017 including 25th-75th percentile range for each size distribution bin. Four seasons are considered: spring (March-May), summer (June-August), autumn (September-November) and winter (December-February).
During spring the median integral concentration $N_{tot}$ is around 1300 cm$^{-3}$. The median size distribution has bimodal shape and is dominated by an Aitken mode centred on 50 nm, and an accumulation mode around 150-200 nm. During the summer

period, the median size distribution is rather mono-modal, and has a less pronounced accumulation mode and a larger Aitken mode around 70 nm. The integral concentration is around 2000 cm$^{-3}$. The autumn period is fairly similar compared to the spring period, although the number concentration is slightly higher (~1900 cm$^{-3}$). The median winter size distribution is distinctly bimodal, with one dominating Aitken mode around 40-50 nm and an accumulation mode around 150-200 nm.

Median integral number concentration is lowest of all seasons, around 900 cm$^{-3}$.

It should be noted that seasonal averages of daily mean aerosol number size distributions do not preserve the signature of new particle formation events (for details regarding new particle formation c.f. e.g. Kulmala et al. 2004). The lack of a distinct nuclei mode in Figure 2 does however not imply that nucleation is absent in the data set, but rather suggest that the intermittent behaviour and short lifetime of the nuclei mode under conditions characteristic for the Aspvreten station leads to

an masking of these features when performing long term averaging. Thus, in Supplementary material, Figure S1, we show Time-of-Day seasonal mean size distributions. As evident, the signature of new particle formation events is present for all seasons except wintertime. 10nm particles are typically observed around noon, but grows rapidly into larger size classes during a couple of hours.

Seasonal averages of the three log-normal modes (referred to as nuclei, Aitken and accumulation mode) fitted to the hourly

size distributions are presented in Table 1. The accumulation mode is largest during summer and smallest during winter, while autumn and spring have approximately same number concentration and modal diameter of the accumulation mode (around 270 cm$^{-3}$ and 160 nm, respectively). The winter period has the largest accumulation mode size (173 nm) and the lowest number concentration (~200 cm$^{-3}$). Aitken mode is largest during summer, both regarding number concentration and size (800 cm$^{-3}$ and 76 nm, respectively). Spring and autumn have quite similar parameters for Aitken mode (average modal

number and size of around 600 cm$^{-3}$ and 60-70 nm, respectively), while winter has the lowest concentration. The nuclei mode is smallest in size, but at the same time largest in number during spring and autumn. This suggests more recent formation or slower growth of newly formed particles compared to the summer period when the size of the nuclei mode is slightly larger. Wintertime data have lowest concentration of nuclei mode particles.

In general, the winter size distribution share features with aged aerosol, and the low number concentration suggest

comparably efficient removal and the distinct minima between Aitken and accumulation mode is indicative of cloud processing (Hoppel and Frick, 1990). Absence of small particles suggest reduced amount of nucleating species likely due to reduction in photochemical activity and/or seasonal changes primary sources. Sulfuric acid vapour has a short residence time and is rapidly consumed by either gas-particle-formation or deposition.

Summertime distributions suggest ageing by condensation growth of both nucleated particles and primary emissions, and

with a lesser influence from cloud processing and wet removal. This is in line with more photochemistry, less frequent precipitation and larger emissions from especially vegetation during the summer months, as pointed out earlier in Tunved et al. (2004). The fact that both spring and autumn have smaller modal size of the Aitken mode suggests less efficient growth by condensation, which may result from less effective photochemistry and lesser source strength of e.g. biogenic volatile organic compounds such as montoterpenes, sesqui-terpenes and isoprene from vegetation. The larger fraction of smaller

nuclei mode particles during these seasons is well in line with studies of nucleation events over the boreal region, that suggest maximum in nucleation occurring during Spring and Autumn (Dal Maso et al., 2005). On average, size distribution properties from the 17-year data record falls in-between those values presented for Hyytiälä and Vavihill (e.g. Tunved et al., 2003;Tunved et al., 2005;Kristensson et al., 2008).

## 3.2 Cluster analysis of size distributions

This section describes the results from the cluster analysis. As stated under Section 3.2, the clustering was performed on hourly means, comprising roughly 130000 size distributions. This approach captures signature size distributions in different stages of the aerosol life-cycle, including aerosol number size distribution types that originate from new particle formation

events (e.g. Kulmala et al. 2004). For each of the twelve clusters (cf. Section 2.3) a number weighted mean diameter was calculated, and the clusters were organized in order of increasing diameter except for what became the starting cluster number one (this cluster stands out by having significantly lower number concentration compared to other clusters). This organization was done to better follow the aging of an aerosol population, where cluster one represents a cyclic point for the aerosol evolution (both a starting and end point). A similar approach was used by Tunved et al., (2004) in order to place the

different clusters in the context of the aerosol life cycle. The median and quartile distributions for each cluster are presented in Figure 3 and the modal parameters are provided in Table 2.

The seasonal and diurnal frequency of occurrence of each cluster is presented in Figure 4 and Figure 5, respectively. Cluster 1 represents the most common aerosol size distribution type, a clearly bimodal size distribution with a dominating Aitken mode. This size distribution is observed about 26 % of the time, and both shape and number concentration suggest extensive

processing by clouds. This cluster is observed mainly during the winter period as can be seen in Figure 4. The average number concentration is about 700 $cm^{-3}$, which is a little more than one third of the annual average concentration (around 1600 $cm^{-3}$). It is interesting to note that observations of size distributions belonging to Cluster 1 exhibit a quite pronounced diurnal pattern (Figure 5), with a maximum around noon. As this cluster is not related to new particle formation (clear absence of freshly formed particles) we hypothesize that cluster 1 have contributions from air entrained from the free

troposphere, resulting in the midday enhancement in occurrence.

While cluster 1 in some respects represents the terminal size distribution, aged under the influence of clouds and precipitation, it also sets the stage for new particles to form due to the low associated condensation sink. In this cycle, cluster 2 represents the youngest form of aerosol size distribution observed. The size distributions belonging to this cluster are evidently shifted towards the smaller size classes with an open-ended size distribution towards particles smaller than

instrumental limit of 10 nm. Cluster 2 has the highest average integral number concentration, around 7500 $cm^{-3}$. The number of larger sized particles is however very small indicating transport from cleaner regions alternatively recent removal by wet deposition. At the same time, this is also the least frequent cluster. The mean diameter of the smallest mode is 15 nm, which also controls the total aerosol number. From Figure 4 and Figure 5 it is also clear that this is mainly a spring and autumn

phenomenon occurring around noon. As the lifetime of such small particles is typically short, this cluster must represent recently formed aerosols. This is consistent with the timing of occurrence, i.e. the time of most efficient photochemical production of aerosol precursors.

Clusters 3, 4, 5, and 6 all share a similar seasonal pattern, with maxima during spring and autumn. The different clusters have varying number concentration, but as pointed out above they are organized according to increasingly larger modal diameter. We understand these clusters as representing different stages of growth following nucleation (i.e. cluster 2). This view is supported by the diurnal distribution of the four clusters; cluster 2 peaks around noon, cluster 3 peaks slightly later, around 15:00 UTC and cluster 4 is shifted yet later into the afternoon. Cluster 5 is mainly observed around 22:00 UTC and cluster 6 is interpreted as growth of the aerosol extending into the following day. New particle formation and subsequent growth observed at boreal sites in northern Europe can typically be traced back in time for about 12 to 24 hours. The peak frequency of occurrence and the increase in modal sizes of cluster's 2-6 are consistent with the evolution observed for individual events. Cluster 2 nuclei mode appears at 15 nm around noon, and cluster 6 is observed 12 hours later with shift in nuclei modal diameter of ~40 nm. This change in size would correspond to an average growth rate of the nuclei mode of about 2 nm per hour, which is in the lower range of reported values from e.g. Hyytiälä (c.f. Kulmala et al., 2004 and references therein).

Clusters 7, 8 and 9 show less distinct features and are more difficult to place in the evolution timeline although the most frequent occurrence of cluster 7 between midnight and 10:00 UTC still would fit growth that extends into the following day. The three clusters do however present larger accumulation mode sizes than clusters 2-6. Moreover, cluster 7 and 9 are mostly encountered during summer, while cluster 8 is observed mainly during spring and autumn. It is also noted that cluster 7 and cluster 9 have higher than average number concentration with cluster 9 having about two times the average number concentrations. We view these cluster as various stages of aged aerosols, but less so than the three last clusters.

Cluster 10,11 and 12 represent the most aged aerosol size distributions with the largest modal size centred around a dominating accumulation mode of about 170-200 nm. These clusters are mainly observed during summer months. Integrated amount of precipitation along the trajectories show that cluster 10 to 12 have the lowest amounts, with cluster 12 having least precipitation of all clusters (data not shown). Thus, these clusters represent aerosol size distributions that undergo ageing by condensation and coagulation with limited removal from the atmosphere. The size distributions in these clusters share features with aged continental aerosol observations, and as they represent the aerosol observations with highest sub-micron mass, we will refer to these clusters as polluted.

Much of the reasoning above is based on knowledge of the tropospheric aerosol knowledge. Some additional support to the analysis was given by the precipitation history of trajectories belonging to the 4 major cluster groups. The precipitation intensity along trajectories belonging to each one of the major cluster groups Washout, Nucleation, Intermediate and Polluted, were averaged and the result was a bar graph was used to illustrate the resulting average precipitation during the last 10 days of transport. It can be clearly seen in Supplementary material Figure S2, that the integrated precipitation is largest for Washout and Nucleation type clusters, and smallest for Intermediate and Polluted. This suggest that the Washout-

type cluster is indeed more likely to have experienced higher precipitation amount en route to Aspvreten. The Nucleation-type clusters exhibit an interesting pattern: precipitation rate is on average high up to some 20h prior arrival where after average intensity decreases. We hypothesize that washout followed by a clearing of skies just before arrival paves way for new particle formation, which in turn highlight the need for both relatively clean air (low condensation sink) together with high photochemical activity to create favourable conditions for new particle formation in the size range of a few nanometres. Hence, the epithet names given to the cluster types is consistent with the expected relation to the evolution of precipitation along the trajectories (Tunved et al. 2004).

Based on the analysis above, we can reduce the clusters into four broader categories. *Washout* (Cluster 1), which characterize winter time CBL aerosol size distributions; *Nucleation* (Clusters 2 through 6), which characterize new particle formation and growth; *Intermediate* (Clusters 7 through 9), which characterize further growth and processing; *Polluted* (Clusters 10 through 12), which characterize summertime aged clusters with low influence of precipitation and strong influence from long range transport of continental aerosol.

We will use these four groupings in linking clusters and their corresponding trajectory source regions. We will also use them in the evaluation of how cluster members have been shifted between the four types during the studied period.

## 3.3    Source regions of dominating cluster types

Dominating transport paths of air mass trajectories belonging to each of the major cluster groups are depicted in Figure 6. As can be seen, the main difference in air mass source regions is found comparing the *polluted* and the *nucleation* cluster groups. *Nucleation* type size distributions are commonly associated with northerly air mass transport between Greenland and Svalbard, while *Polluted* type cluster show pronounced south-westerly transport on average, with major pathways over Great Britain and northern parts of continental Europe. This is in agreement with previous findings, e.g. (Tunved et al., 2003;Tunved et al., 2005;Sogacheva et al., 2005;Sogacheva et al., 2008), showing that nucleation observed over the Nordic region predominantly occurs in clean marine Arctic air masses. The transport characteristics for the *Intermediate* type size distribution clusters show resemblance to the transport pattern observed for the *nucleation* type clusters, suggesting this type of size distributions indeed fit the picture of an aged nuclei mode resulting from nucleation taking place upstream of the receptor. The *washout* cluster type shares source regions with the *polluted* type cluster, but with a high degree of wet removal masking the influence of sources apparent in the *polluted* cluster type.

# 4. Decadal trends in aerosol number size distribution properties

## 4.1 General trends of the aerosol size distribution

Figure 7 shows the estimated annual linear trend (and 95%CI) for each one of 16 size bins expressed as Theil-Sen estimator slope together with the grand average of the number size distribution with 25th-75th percentile ranges indicated. The Theil-Sen estimator represents the median of all slopes derived from pairs of values $(i, j)$ from the dataset, $(y_j - y_i)/(x_j - x_i)$, and constitutes a robust measure of trends for data series that are not normal distributed. For the statistical calculations we have made use of the ktaub-statistical Matlab package (Burkley, 2006) and references therein. Prior trend analysis, all size distribution data was remapped to 16 size bins over the observed size range, using a bin width of dlogDp equal to 0.1. The trend is expressed as the change in size distribution as d(dN/dlogD$_p$) per annum. It is from Figure 7 clear that a negative trend is present for all size classes. The nuclei mode particles exhibit only a modest downward trend (>-5 cm$^{-3}$ y$^{-1}$) and remains about same up to 40 nm. Then the decreasing trend gets gradually larger with increasing size, reaching peak values of -35 cm$^{-3}$ y$^{-1}$ around 80 nm. Hereafter the negative trend decreases in magnitude up to around 120 nm. Between 120 and 200nm, the negative trend again gets larger, reaching a second mode peaking at a downward trend of -28 cm$^{-3}$ y$^{-1}$. After this peak, the negative trend decreases with increasing size to the upper limit of the instrument at 400 nm. It is noted from Figure 7 that the trend does not correlate with the average concentration of particles. Figure 7 also suggest two modes of the decreasing trend, one around 80 nm and one centred at 200 nm. The gradient in trend is further very sharp between 40 and 80 nm. The presence of the two modes could be suggestive of mainly two different processes acting independent on the size distribution. The presence of the two modes could be suggestive of two different processes acting independent on the size distribution. Sulfur emissions in Europe have seen a dramatic decrease during the last decades ((Smith et al., 2011;Aas et al., 2019). Being one of the prime precursors for formation of aerosol number and mass formation, we attribute the strong negative trends around 80 and 200nm to a reduction in aerosol sulfate. The first mode likely reflects the reduced gas-particle-production occurring far from the receptor through condensation of sulfates. The second mode however is likely the result of less in cloud oxidation of SO$_2$, a well-known production pathway for accumulation mode sulfate mass (e.g. Langner et al., 1992). The lack of any major trends present for particles smaller than 40 nm suggest that the period 2000-2017 have seen little changes in particle production close to the site. The two modes at 80 nm and 200 nm are however consistent with changes in processes occurring far away from the measurement site in air masses where the aerosol have had time to age from both condensation and in-cloud processes.

The trend analysis was further extended to investigate if any seasonal variation could be found in the size dependent trends. Figure 8 shows the seasonal size resolved trend for the period. As can be seen a rather complex picture emerges, where trends vary with both size and season. Some generalizations can however be done. During December, January and February all size classes show a decreasing trend. This is followed by the most substantial negative trend, found in March through May for size range 50-300 nm. The trend, d(dN/dlogD$_p$) per annum, is in excess of -40 cm$^{-3}$ y$^{-1}$. During the same time of the year there is a concurrent moderate increasing trend for particles smaller than about 40 nm (~10 cm$^{-3}$ y$^{-1}$). During the

summer period between mid-May and August, the negative trend for particles larger than about 70 nm is less distinct compared to the spring period. However, particles 25-70 nm present a strong positive trend and this trend is similar in magnitude but opposite in sign compared to the accumulation mode trend in spring (-30 cm$^{-3}$ y$^{-1}$). In autumn the positive trend for the small particles is abruptly changed to a negative trend, and essentially the whole aerosol size range display decreasing concentration with time. The negative trend is strong in the size range about 40 and 70 nm (around -40 cm$^{-3}$ y$^{-1}$). Evidently, strong gradients exist in estimates of trends over the observed size range and season, and same sized aerosols can show both positive and negative trends, depending on season. In figure S4, 95% Confidence interval of Theil-Sens slopes for lower (left) and upper (right) confidence interval. Color indicate calculated linear trend for binned particle number concentration at Aspvreten as particle cm$^{-3}$ year$^{-1}$ for the time period 2000-2017. Areas bounded by the dashed red line represents pairs of month/size bin where test for significance was below the 95% threshold.

## 4.2    Trends of fitted size distribution properties

In order to attribute the trend information depicted in Figure 8 to the different modes described in 2.2, trends were derived for each mode, now on annual basis. From the information in Figure 8, we are aware that the nuclei and Aitken mode will be a mix of positive and negative trends on annual basis. Nevertheless, this procedure will help in establishing a confidence in the observed trends over the data set as a general feature rather than a step change due to changes in infrastructure.

Figure 9 shows the annual medians of modal diameter (Figure 9 a-c) in micrometres for mode 1-3, and the corresponding modal number concentration (Figure 9 d-f) in cm$^{-3}$. Also indicated in the figure title is the Theil-Sen estimator of the slope of the geometric mean diameter and modal number concentration. All trends were tested using the seasonal Kendall test on the hourly data (95%CI), and Table 3 summarize the results from calculations of Theil-Sen estimator slopes together with confidence intervals of derived slopes for whole year.

As can be seen in Table 3, the median trend for the nuclei mode on annual basis is -10.9 cm$^{-3}$ y$^{-1}$ (95%CI -8.5 to -12.2 cm$^{-3}$ y$^{-1}$). Aitken mode number concentration trend was found to be -9.2 cm$^{-3}$ y$^{-1}$ (95%CI: -7.1 to -11.4 cm$^{-3}$ y$^{-1}$) and accumulation mode concentration trend was calculated to -8.2 cm$^{-3}$ y$^{-1}$ (95%CI: -7.4 to -9.1 cm$^{-3}$ y$^{-1}$). All trends of number concentrations were found significant when applying seasonal Kendall test following the method proposed by (Hirsch and Slack, 1984). The choice of seasonal Kendall test was motivated by the serial dependencies apparently present in the data set (i.e. seasonal variation). Upper and lower 95% confidence intervals of slopes were calculated following Hollander and Wolfe (1973).

Regarding the trends of modal geometric diameters, calculations of the Theil-Sen estimator slopes gave only a slight negative trend for the nuclei mode of -0.07 nm y$^{-1}$ (95%CI: -0. 00 to -0.14 nm y$^{-1}$) and a little stronger decrease for the Aitken mode geometric mean diameter (-0.31 nm y$^{-1}$, 95%CI: -0.20 to -0.44 nm y$^{-1}$). Most marked decrease was found for the accumulation mode geometric mean diameter. The average trend over the studied period was found to be -2.01 nm y$^{-1}$ (95%CI: -1.76 to -2.24 nm y$^{-1}$). This means that since observations started in 2000, the annual mean modal size of the accumulation mode has decreased by almost 40 nm, conjunct with a number decrease of around 150 cm$^{-3}$. These numeric

values are very large compared to the mean seasonal values listed in Table 1, and indicative of major changes in both size and number concentration of the accumulation mode.

A seasonal trend analysis for the modal number concentration is presented in Figure 10 and a similar seasonal variation as depicted in Figure 7 is apparent. The accumulation mode is negative for all months with the strongest decrease present during April, -13 cm$^{-3}$ y$^{-1}$. Over the rest of the year the decrease is in the range between -5 to -10 cm$^{-3}$ y$^{-1}$. Numerical values for each mode and month are summarized in Table 4.

Different from the accumulation mode, the nuclei and the Aitken mode trends are weakly positive in June and July. From a positive trend in summer, the nuclei mode presents the overall strongest monthly decreasing trend of the three modes in October, with about -20 cm$^{-3}$ y$^{-1}$. Except for the months of June, July, and November regarding the Aitken mode, all other trends in Figure 10 and Table 4 are significant according to the Mann-Kendall test at 95 % confidence level.

As an additional analysis, the temporal evolution of aerosol submicron mass was investigated. The aerosol number size distribution was recalculated to mass size distribution assuming a density of 1 g cm$^{-3}$, followed by integration over the instrumental size range. In this way mass was calculated as daily means (µg m$^{-3}$) for the period 2000-2017. Theil-Sen estimator slope was calculated for this dataset. Figure 11 presents the result of annual mass concentration as a Box-Whisker plot and estimated Sens slope. As can be seen, the reduction of mass follows the general trend of aerosol accumulation mode number concentration closely. A Theil-Sen slope of 0.077 µg m$^{-3}$ y$^{-1}$ (95%CI -0.085 to -0.071 µg m$^{-3}$ y$^{-1}$) was estimated. Based on this calculation, we estimate that mass was reduced by roughly 52 % during the studied period. Torseth et al. (2012) estimated an average decrease of PM2.5 of 27% 2000-2009 for several European stations. In Figure 5 and Figure 6 of Torseth et al. (2012), average PM2.5 mass and an estimated trend is visible for Aspvreten. PM2.5 average concentration at Aspvreten is in the range of 4-7 µg m$^{-3}$, and our calculated mass (assuming spherical particles and unit density) is about 2.2 µg m$^{-3}$. This suggests that at least 50 % of PM2.5 is controlled by particles smaller than approximately half a micron (this is a lower range estimate since we use density of 1 g cm$^{-3}$, so doubling our assumption of density will put sub-390 nm mass in range of 4.5 µg m$^{-3}$). Our calculated decrease in mass 2000-2009 is roughly 30 %, i.e. slightly less decrease of PM2.5 than estimated by (Torseth et al., 2012, in excess of 40 %). This suggests that the negative trend of particles >390 nm is slightly larger for particles above the size range covered by our instrument. Nevertheless, our estimated trend is still in good agreement with observed PM2.5 trends and it is evident that a major part of PM2.5 reductions is controlled by particles smaller than 400 nm. Furthermore, Glantz et al. (2019), found a significant decrease in Aerosol Optical Thickness (AOT) over the western Gotland basin area of 1.5%, 1.1% and 1.6% per annum derived from MODIS c051, MODIS c061 and AERONET GDT, respectively. This would translate to an overall decrease of about 25% over the 17-year period covered by our study, which agrees well with our derived submicron aerosol mass decrease of ~30%.

## 4.3 Trends in cluster groups

Each size distribution observed at the station is attributed one of the 12 cluster types and further grouped into one of four categories (*Washout, Nucleation, Intermediate, and Polluted*), as described in section 3.2. For each year, the sum of the different categories represents 100 %. If the relative contribution of a category remains the same over the time period, there will be no significant trend for that category. If a category increases or decreases significantly, this must be reflected in an opposite change in one or more of the other categories. The trend analysis of the cluster groups is presented in Figure 12.

The two clearest changes over time is the increase in *Washout* (cluster 1) and a corresponding decrease in *Polluted* (clusters 10-12), both at a rate of about 1 % units per year. A closer look at the polluted category shows that cluster 12 almost completely disappears over the measurement period. The two other categories show smaller changes. The *Nucleation* presents a small decreasing trend of 0.1 % units per year, and the I*ntermediate* category show an increase of about the same magnitude, 0.2 % units per year.

Whereas the *washout* category almost doubled over the measuring period, this was almost entirely compensated by the decrease in the *Polluted* category. These changes combined could conceivably lead to a lowering of the condensational sink (CS) by the aerosol (cf. Table 3) and this would tentatively promote new particle formation. However, based on the data from Aspvreten, the negative trend in the category *Nucleation* does not support a simple direct link between CS and new particle formation at the station. The slight increase in the *Intermediate* category could indicate that if there were still a small enhancement in new particle formation, but that it must have occurred at some distance away from Aspvreten and had time to age several days.

## 4.4 Trends in potential CCN concentration

In order to study the potential effect of the observed changes presented above on availability of cloud condensation nuclei (CCN), each observed size distribution was used to initialize a simple adiabatic cloud parcel model. The model, which constitute a module in the aerosol transport model CALM (Tunved et al., 2010), calculates cloud droplet activation in warm clouds (i.e. without ice component) under adiabatic conditions.

This is a highly generalized assumption that neglects the influence of competitive growth in mixed phase clouds. The introduction of ice crystals into a pure liquid phase cloud causes the ice crystals to grow on expense of the liquid phase due to the relatively lower saturation vapour pressure over ice compared to liquid droplets. In clouds that are warmer than -36C, formation of ice crystal occurs on so called ice nuclei, typically mineral dust and soot particles of larger diameters (DeMott et al., 2010) although other particle types may serve as ice nuclei as well (e.g. Murray et al., 2012). In Kanitz et al. (2011) Figure 3 presents observed fraction of ice-containing clouds in different environments function of cloud top temperature. As can be seen, there is a high degree of variation in fraction of ice containing clouds depending on environment studied. For

e.g. the Leipzig dataset, 70% of the clouds contain ice at -19°C, while similar ice fraction is only achieved at -34°C in the Punta Arenas dataset.

If we assume our calculated estimate is restricted to low level clouds, and given that the average surface temperature at Oxelösund (a few km south-west of Aspvreten) are below zero degrees Celsius during winter months only (Data source: Surface temperature, hourly values, from Swedish Meteorological and Hydrological Institute, SMHI, www.smhi.se, 1961-2018), the problem arising from mixed phase clouds is likely contained to the winter months December-January. Thus, our assumption regarding liquid phase only cloud droplet activation and growth is likely prone to errors during this season and is likely to overestimate the CCN number in our idealized simulations.

A simplified chemical system of internally mixed aerosols was assumed, consisting of ammonium bisulfate and some hypothetical organic compound with a solubility of 10 % and a Van't Hoff factor of 1. The Van't Hoff factor for $NH_4HSO_4$ is set to 2.4, and the solubility is assumed to 100 %. Water accommodation coefficient is set to 1.

The evolution of the droplet spectra for each cloud simulation is calculated until peak super-saturation (S) is reached in the parcel, i.e. for as long as $dS/dt > 0$, to assure that no more activation can take place. Once calculations are terminated, the number of activated particles is integrated and stored as the CCN value for that particular size distribution. This approach will allow for a slight variation in the altitude where peak super-saturation is reached, and this in turn depends on updraft velocity and chemical makeup.

Each aerosol size distribution was tested using three different chemical characteristics based on $NH_4HSO_4$:ORGANIC ratios of 50:50, 90:10 and 10:90. For each chemical mixture three different updraft velocities were used, 0.1 ms$^{-1}$, 0.5 ms$^{-1}$, and 1 ms$^{-1}$, respectively. The nine different scenarios are summarized in Table 5.

The annual median CCN resulting for the 50:50 mixture and three different updrafts are presented in Figure 13. All three updrafts give a similar impression where the CCN drops decreased by more than 1/3 of the original concentration during the observation period. The linear trends, calculated as Theil-Sen slopes, provide numerical values of about -5 cm$^{-3}$ y$^{-1}$, -10 cm$^{-3}$ y$^{-1}$, and -13 cm$^{-3}$ y$^{-1}$ for the low, medium, and high updraft scenarios. The two other cases, with 90:10 and 10:90 mixtures, follow the negative trend as presented in Figure 13, but with the difference of being parallel shifted. That is, stronger trends for the more soluble mixtures as opposed to weaker trends for the less soluble scenarios. All the nine slopes are summarized in Table 5. The difference in trend is largest for the high updraft scenarios comparing the different solubility assumptions. The trends are similar for the low updraft case, -5.0 (-5 to-5), -4.3(-4.8 to -3.9) and -3.5(-3.8 to -3.2) cm$^{-3}$ y$^{-1}$ for compositions 50:50, 90:10 and 10:90, respectively. For the high updraft case corresponding values are -13.3(-15.0 to -11.7), -15.0(-17.0 to -13.0) and 9.3(-10.3 to -8.4) cm$^{-3}$ y$^{-1}$.

It is possible to make simple estimates of changes in cloud optical thicknesses based on these trends. If the cloud water content in our clouds are unaffected by the decreasing trend in CCN, we can simply write

$$\frac{\Delta \tau}{\tau} = -\frac{1}{3}\frac{\Delta N}{N}$$  (3)

where the term on the left is the relative change in cloud optical thickness and the term to the right is our estimated relative change in CCN based on the analysis above divided by 3. The relative changes in cloud optical thicknesses are summarized in Table 5.

For all our scenarios, the expected relative increase in cloud optical thickness is greater than 9 %. The largest relative change in cloud albedo of 16 % between 2000 and 2017, occurs for the assumption of least soluble particles and the smallest updraft velocity. The high sensitivity for these conditions is expected, since both low updraft and less soluble particles will generate few cloud droplets and thus high sensitivity to even moderate changes in cloud droplet number concentration, all in agreement with (Twomey, 1991;Platnick and Twomey, 1994). If we assume that the most realistic composition is somewhere between the 50:50 and 90:10 chemical makeup, this would yield a cloud albedo change in the range of 10-12 %.

The annual trends provide a very robust picture of a decrease in potential CCN and the likely decrease that follows in cloud optical thickness. However, our analysis above also emphasizes that the calculated trends in size distribution properties are not the same over the whole year. We therefore repeat or model calculations on a monthly basis. Figure 14 combines all the nine combinations of updraft and chemical mixture and the resulting trends in CCN as function of month.

Figure 14 points to some interesting general features. Firstly, the strongest contributors to the overall decrease in CCN are March and April. The strong decrease in accumulation mode particles cannot be compensated by the small local enhancement in nuclei mode (cf. Figure 18). Secondly, despite the summer increasing trend in nuclei and Aitken modes, the net effect on CCN is negative. Thirdly, the strongest decreasing trend presented by the nucleation mode in September and October, is not reflected in the trend of CCN in the same period, suggesting that CCN is controlled by larger particles transported to Aspvreten.

Overall, it is clear that a substantial reduction in CCN number is likely to have taken place during the studied period. In fact, if we assume the scenario 50:50, with an average updraft of 0.5 ms$^{-1}$, the 17-year period has brought about a change in CCN as much as 360 cm$^{-3}$ in the month of April. Of course, other changes different from the decreasing trend in aerosol concentration may also have taken place, which could have altered the potential availability of CCN.

## 4.5    Trends in general transport patterns

The indisputably and generally significant trends in the aerosol size distribution on annual and monthly basis throughout the measurement period are expected as a response to a reduction in anthropogenic emissions of aerosols and their precursor gases, mainly sulfur dioxide. However, these analyses must be complemented with an analysis of possible variations in transport patterns over the same time period. To assess the contribution of observed trends from changes in air mass

transport patterns, a polar coordinate grid system was arranged around Aspvreten spaced in 2.5-degree increments in the lateral and 10 degrees in the longitudinal direction.

Each occasion a trajectory passed over a grid box was recorded as an event. For each year, the sum of all used trajectories for that year was used to normalize each of the grid boxes. This results in a new value associated with the grid box representing a yearly relative probability that a trajectory passed a given grid box en route to Aspvreten (compare with Figure 6). For each grid box a performed a first order fit to represent the trend using polyval.m function in Matlab. The result is presented in Figure 15.

The figure shows the sign (positive or negative) and magnitude of the trend of the probability of a random sampled trajectory to cross a given grid cell. A positive trend thus means that more trajectories are crossing that grid cell today compared to 17 years ago. All in all, given the length of the trajectories this reflects changes in the Meso-scale circulation.

As can be seen in Figure 15, the likelihood of direct northerly transport has decreased, while the south-westerly component has increased, resulting in more frequent air mass transport from North Atlantic region including areas around Greenland and Iceland. There is only very minor reduction in the amount of air transported from continental regions.

Albeit the systematic change in the transport pattern presented in Figure 15, the trends are small in comparison to the large changes in aerosol properties over the same time period. We cannot however completely exclude any possibility of transport related changes in observed aerosol concentrations at Aspvreten. We do however believe that contributions from observed small changes in transport patterns to the very large observed trends are small.

We further attempted to relate the increased relative frequency of Washout-type clusters to trends in precipitation amount along the trajectories. The analysis did not result in any clear time dependent trends. In order further investigate the relative roles of wet removal and source strength, we revisited the data and made an analysis of average precipitation intensity along Cluster 1 (i.e. Washout-type cluster), creating hourly resolved bar-graphs for pre-2009 and post-2008 periods, to see if there exist any differences. If post 2008 data contain trajectories with on average more precipitation, enhanced wet removal may play an important role. It is clear, that integral precipitation on average is about the same for Cluster 1, but slightly different patterns in timing of precipitation can be noted. This tells us that at least for the last ~2 days, we have, if anything less precipitation influence. This in turn do suggest that the relative increase of size distribution observations belonging to Cluster 1 is not likely to be the result of increase in precipitation, and if we assume that the sink strength would be the same, the redistribution of cluster members in favour of Cluster 1 must be dominated by  emission reductions and not enhanced sink processes.

It should also be added, that there is a substantial uncertainty in calculation of trajectories, especially in regions where possibilities of meteorological model validation are poor like in the Arctic region. The results presented here do however indicate that if changes in transport pattern in fact take place, the changes over the studied period seems small, or non-linear.

## 5.    Discussion

In the presented study we have made use of an almost 20 years of continuous dataset comprising aerosol number size distribution data between 10-390 nm. Data availability during the studied period is above 80 %. The data has been analysed from several different aspects in order to shed light on the degree, when and where changes in emissions may have had an impact on the observed aerosol size distribution. Results show generally negative trends for annual mean concentrations and sizes of the three modes, referred to as nuclei, Aitken and accumulation mode. Based on the combined cluster and air-mass analysis, this decreasing trend, in the order of 10 particles per mode and year, is consistent with reduced emissions of sulfur dioxide and other anthropogenic primary particles as well as aerosol precursor gases. Aerosol integral number have been reduced by roughly 30 %, from ca 1800 $cm^{-3}$ in the beginning of the period to about 1300 $cm^{-3}$ at the end of the studied period. The general trend found in this study is well in agreement with the findings presented by (Asmi et al., 2013), and extrapolation of their derived trends (N20, Asmi et al., 2013, Table 2) over the same time period presented in this study would give results in the range of  23%, 20% and 40% at Vavihill, Hyytiälä and Värriö and an increase of 2% at Pallas. A simple estimate of mass trends based on the presented trends for N100 yield values similar to our estimates of sub-micron mass trends found in the current study. In addition, cluster analysis of hourly number size distributions has been demonstrated to be a useful tool in trend studies. The method has been applied to study how the aerosol observations are distributed over 12 signature distributions, and further applied to investigate how the aerosol have been re-distributed between these 12 dominating cluster types during the period 2000-2017. The method seems well suited for studying trends in new particle formation events. It is however clear that there is more to the story than simply an overall decrease over all seasons and over the entire size range. We have shown that the trends can be negative or positive depending on what season and over which size range the trend is calculated. This is indicative of the complex interplay between aerosol sources and sinks, their seasonal variation as well as the dynamics of new particle formation.

In the case of the accumulation mode, fitted modal diameter has decreased by 2 nm $y^{-1}$ (in total, a 17 % decrease or ~35 nm absolute decrease, c.f. Figure 9). This decrease, together with the accumulation mode number decrease, has been associated with a major reduction in submicron mass. We show that integrated particle mass 10-390 nm have decreased by more than 50 % during the studied period (as annual averages, c.f. Figure 11). This decrease is well in agreement with observed trends of PM2.5. We further show that a substantial part of PM2.5 mass is made up by particles smaller than 400 nm. This in turn suggests that observed PM2.5 changes to a large degree are controlled by particles smaller than about 0.5 micron. We also show results suggesting that two main sulfate production pathways have decreased in magnitude, namely gas phase oxidation followed by condensation and wet phase sulfate production in non-precipitating clouds.

More complex is the behaviour of Aitken and nuclei modes. Since the nuclei mode reflects recent formation of particles, the general reduction in nucleation mode particles must reflect reduced particle formation in the relative vicinity of the station. However, as particle formation is largely controlled by the condensation sink (CS) of pre-existing particles (Boy et al., 2005;Dal Maso et al., 2008;Kulmala et al., 2001), and the accumulation mode is mainly controlling CS, a reduction in

accumulation mode number and size will also decrease the condensation sink. Therefore, one would expect a reduction in CS to promote new particle formation. As shown by the calculated trends, a general increase in nuclei mode is not observed. This indicates that the reduction in CS is accompanied by a concurrent reduction of precursors available to nucleate new particles, which counterbalance the effect of a reduced CS. As sulfuric acid is one of the main key candidates for new particle formation, we conclude that the reductions of sulfur emissions thus feedback through both reduced CS and reduced nucleation. Moreover, there is likely no other nucleating species that appears to have replaced sulfuric acid, with respect to new particle formation in the studied region. In that case, we would have expected a stronger response in the nuclei mode due to the strong changes in CS. Another possibility that we cannot rule out is that changes in precipitation during the studied period have resulted in more efficient removal of both accumulation mode particles and nucleating species. Recalling Figure S3 in Supplementary material, any trend as indicated by trajectory derived precipitation includes a decrease or redistribution along the transport paths comparing pre-2009 to post 2008 periods. As we have not explicitly studied cloudiness over the site, we can neither rule out the possibility that on average increased cloudiness have reduced the photolysis rate close to Aspvreten. Both the role of precipitation and cloudiness in changes of new particle formation would likely deserve a dedicated study. In figure 8 we show that a positive trend for Aitken mode size particles is present for the summer months June-July. This increase may reflect improved conditions from a new particle formation perspective, and this may reflect favourable relation between $H_2SO_4$ and CS or some other nucleating species. As this increasing trend is absent for the smallest size classes, we conclude that if this increase is the result of enhanced nucleation, we argue that it must take place upstream of the station rather than in the immediate vicinity in order to reach the size range where we observe the positive trend. In essence; nucleation occurs earlier during the marine-continental transition and this in turn quench further nucleation closer to the station. We have also performed simulations to assess the potential influence of observed trends on cloud droplet number concentration. We show that a negative trend is present for all months, but the calculated decrease is largest during spring. The springtime reduction ranges from -5 $cm^{-3}$ $y^{-1}$ to ~-30 $cm^{-3}$ $y^{-1}$ depending on assumption regarding updraft and chemical composition. We estimate that the reduction of potential CCN's corresponds to a reduction in cloud albedo of about 10 %.

We also studied the potential influence on observed trends resulting from changes in transport pattern. We do however conclude that these are small and likely of less importance for the observed reduction in particle number and mass. Instead we suggest that emission reductions of anthropogenic emissions, notably $SO_2$, are the main driving candidate for the observed change. These changes have the largest effect during the spring, and this period is most sensitive to changes in anthropogenic emissions in absolute terms. Further, the sensitivity is also largest for the accumulation mode number concentration and size, which in turn represents the most efficient light scattering and cloud forming aerosol particles.

## 6.    Summary and Conclusions

In summary, our main findings are presented as bullet points below:

As revealed by the aerosol size distribution clustering, it is evident that the cluster representing clean, cloud processed aerosol is increasing on expense of the polluted type monomodal size distribution. There is only marginal increasing trend of cluster members belonging to clusters showing sign of recent new particle formation. At the same time, the most polluted cluster 12 has been reduced from around 5% of observations to around 1% during the period 2000-2017. We have shown that cluster analysis successfully can be used to study aerosol trends, and further that cluster analysis also can be used for studying trends of intermittent processes such as new particle formation events. The method has clear advantages compared to standard time averaging techniques as it preserves the shape and number concentration of the aerosol number size distributions within the clusters which otherwise easily can get lost in usually applied time averaging such as mean and medians.

- There is on average a negative trend of 10 particles $cm^{-3} y^{-1}$ for nuclei, Aitken and accumulation mode. During the 2000-2017 period, annual average integral number concentration has been reduced by 20-30 %.

- Integral number concentration has been reduced from about 1800 $cm^{-3}$ to about 1300 $cm^{-3}$, or by almost 30 %

- Annual mean accumulation mode size has decreased by 35 nm or 17 % during the 2000-2017 period. This suggests that the reduced emissions on result not just in fewer particles, but also that they are, on average, smaller.

- Aerosol mass (10-390 nm) have decreased by more than 50 % during the 2000-2017 period. This trend is found to be in good agreement with trends derived for PM2.5. The results further suggest that a large part of previously presented PM2.5 trends are controlled by particles smaller than 0.5 micron.

- There is a pronounced seasonality in the trends of aerosol number concentration. The strongest reduction is found during spring in the accumulation mode, and during autumn for the Aitken mode. In spring, the annual trend in accumulation mode number concentration is found to be close to 15 $cm^{-3} y^{-1}$ and the reduction in nuclei mode concentration during autumn is calculated to around 20 $cm^{-3} y^{-1}$ during September and October.

- Although we do not find evidence for increased nucleation close to the measurement site, we do find a positive trend for Aitken mode sized particles during summer months. Since the accumulation mode concentration is reducing, we interpret the increase in Aitken particle concentrations as a response to the decrease in condensation sink, leading to more frequent nucleation upstream of the station (c.f. Section 5.).

- A calculation of potential CCN using an adiabatic cloud parcel model for various assumptions regarding updraft velocity and chemical composition showed that the number of CCN have likely been substantially reduced during the 2000-2017 period. This in turn yields as a best guess about 10-12 % decrease in cloud optical thickness. The negative trend is present for all months and for all assumptions regarding updraft and chemical composition, but the largest negative trend is found during spring, ranging from -5 to almost -35 CCN $cm^{-3}$ $y^{-1}$, depending on assumptions applied regarding chemical composition and updraft velocity.

- We suggest that main drivers for the observed reduction in aerosol number and mass are reductions of anthropogenic emissions in general and reduced $SO_2$ emissions in particular.

*Author contributions* Idea for manuscript and data analysis routines were developed by Peter Tunved. Data interpretation, literature search and writing were performed by Peter Tunved and Johan Ström.

*Competing interests* The authors declare that they have no conflict of interest.

*Acknowledgements.* Funding for this study was provided through the Swedish Environmental Protection Agency (Naturvårdsverket,NV) as part of the environmental monitoring program. Hans Karlsson is greatly acknowledged for work and maintenance of the aerosol instrumentation at Aspvreten.
We thank the two anonymous reviewers for their constructive comments. The suggested revisions and additions made the manuscript clearer enhanced the overall quality of the manuscript.

*Data used* The data used in this study has been published in PANGAEA data repository reference
https://doi.pangaea.de/10.1594/PANGAEA.900502

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

**Figures and captions**

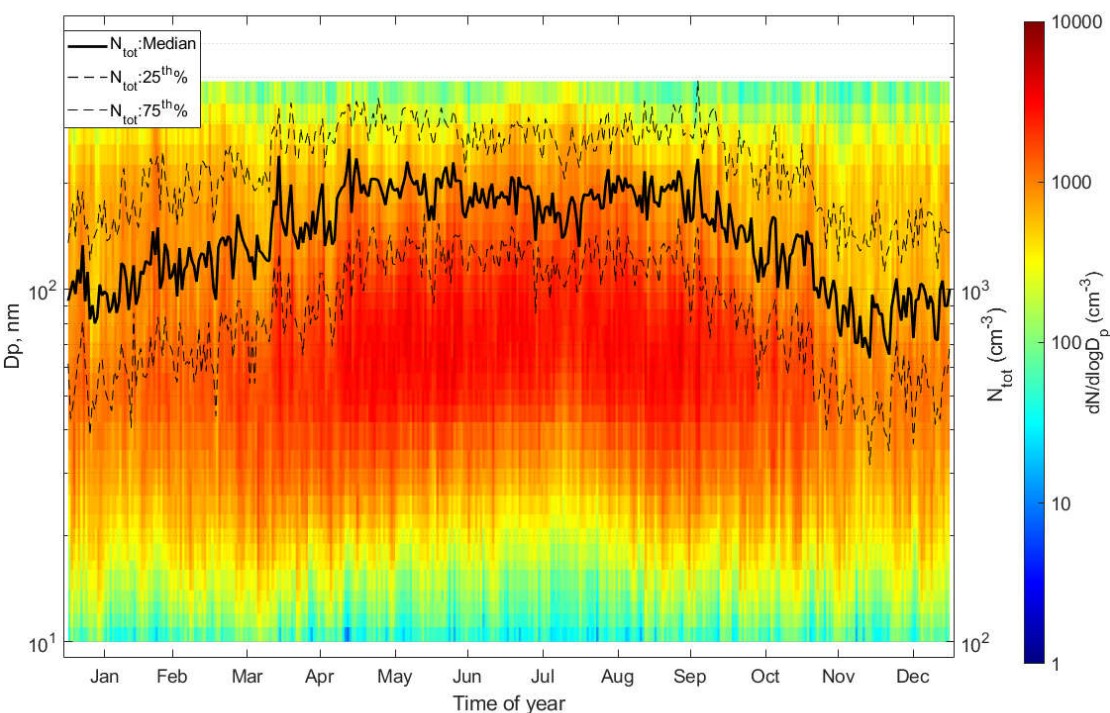

**Figure 1: Seasonal variation of the aerosol number size distribution between 10-390nm presented as daily median aerosol number size distribution for the whole study period, 2000-2017. Superimposed on the surface plot is the median and quartile ranges of integral number concentration.**

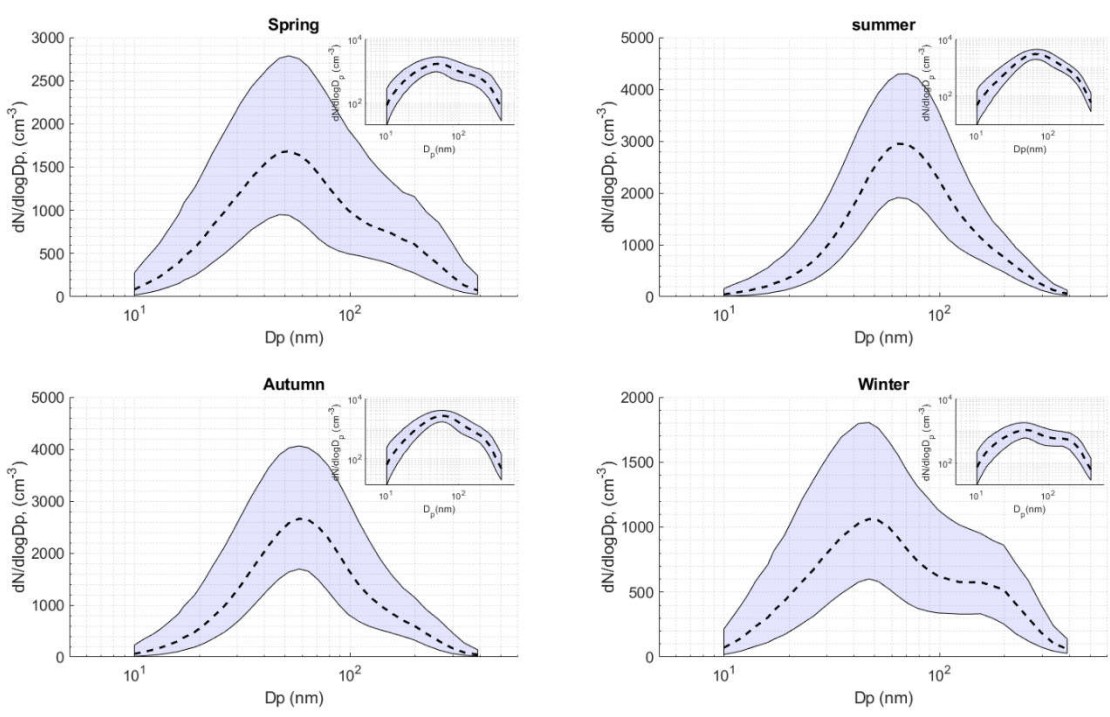

Figure 2: Seasonally average size distributions observed at Aspvreten 2000-2017. Shaded blue area gives 25th-75th percentile ranges, and dashed line median size distribution. Shown in the sub-frames are same data, but now in log-log scale. Spring=March-May; Summer=June-August; Autumn=September-November; Winter=December-February.

| Spring | N [cm$^{-3}$] | 25th-75th% | GSD | 25th-75th% | Dg[nm] | 25th-75th% |
|---|---|---|---|---|---|---|
| Nuclei | 618 | (251-1252) | 1.5 | (1.33-1.715) | 33 | (21-48) |
| Aitken | 581 | (187-1271) | 1.49 | (1.3-1.845) | 67 | (47-99) |
| Accumulation | 277 | (127-568) | 1.44 | (1.34-1.638) | 164 | (114-211) |
| **Summer** | | | | | | |
| Nuclei | 573 | (231-1185) | 1.45 | (1.3-1.677) | 41 | (24-58) |
| Aitken | 800 | (344-1414) | 1.38 | (1.25-1.74) | 76 | (61-93) |
| Accumulation | 375 | (173-803) | 1.41 | (1.29-1.595) | 153 | (108-199) |
| **Autumn** | | | | | | |
| Nuclei | 661 | (276-1300) | 1.5 | (1.34-1.72) | 35 | (22-49) |
| Aitken | 644 | (230-1264) | 1.45 | (1.28-1.792) | 67 | (49-91) |
| Accumulation | 262 | (126-550) | 1.4 | (1.3-1.555) | 159 | (115-202) |
| **Winter** | | | | | | |
| Nuclei | 372 | (155-762) | 1.57 | (1.39-1.786) | 33 | (21-46) |
| Aitken | 297 | (101-684) | 1.52 | (1.29-1.912) | 73 | (49-115) |
| Accumulation | 199 | (95-371) | 1.43 | (1.32-1.586) | 173 | (134-212) |

**Table 1: Statistics of modal fits per season. Table shows statistics derived from fitted hourly number size distributions. Indicated in table are Nuclei, Aitken and accumulation mode parameters as median and 25th-75th percentile. GSD represents the geometric standard deviation and Dg the geometric mean diameter of each one of the log-normal modes.**

range        and        median        of        *all*        data,        for        reference.

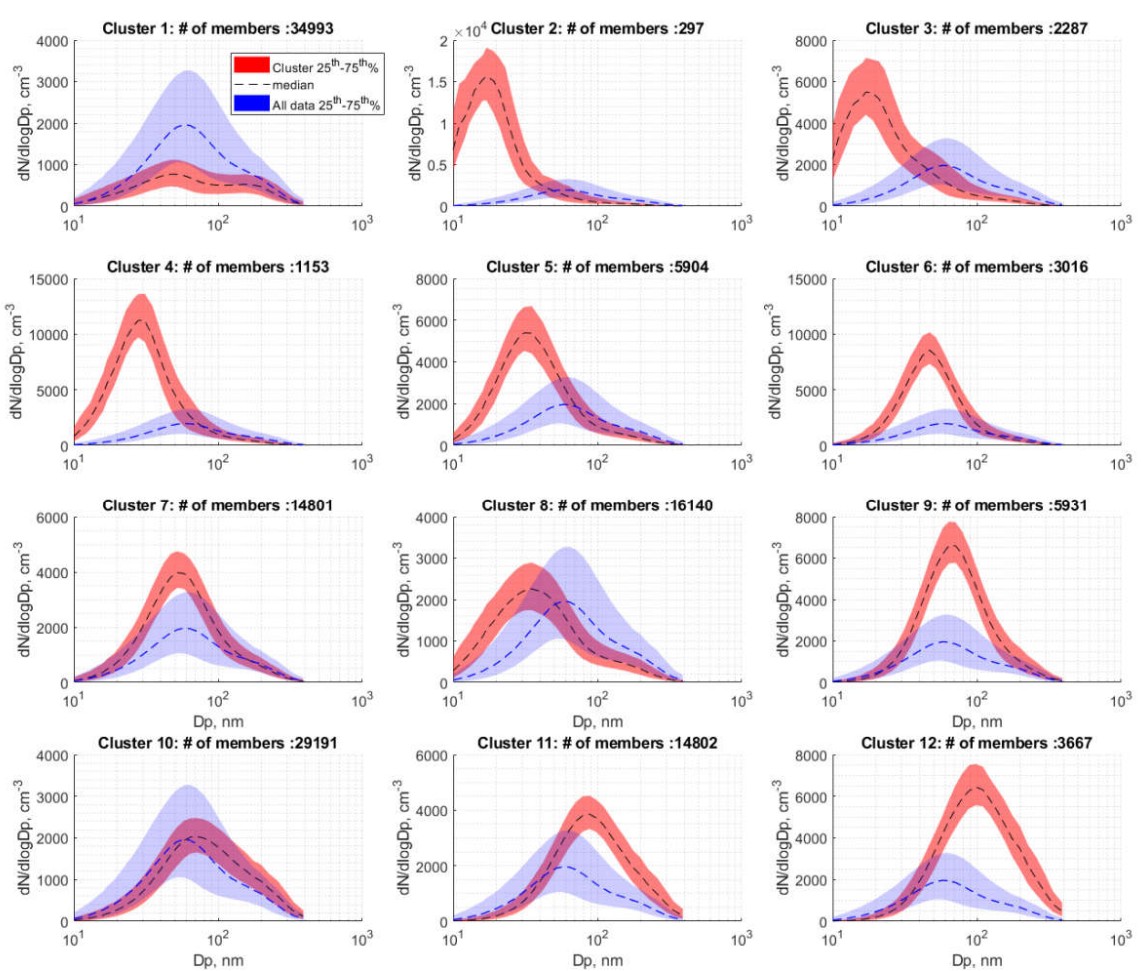

**Figure 3: Aerosol number size distribution clusters for the period 2000-2017 data record from Aspvreten, 58.8N, 17.4E. a total of 132182 hourly size distributions included in the clustering. Number of cluster members is indicated in each frame. Clusters are ordered left→right, up→down. Red shaded area indicates the 25th-75th percentile range of the cluster members and b dashed line indicate median number size distribution of that cluster. Indicated in blue surface is 25th-75th percentile. GSD represents the geometric standard deviation and Dg the geometric mean diameter of each one of the log-normal modes.**

| Cluster id | Mode | N (cm$^{-3}$) | 25$^{th}$-75$^{th}$% | GSD | 25$^{th}$-75$^{th}$% | Dg(nm) | 25$^{th}$-75$^{th}$% |
|---|---|---|---|---|---|---|---|
| **Cluster 1** | *Mode 1* | 291 | (136-509) | 1.58 | (1.39-1.815) | 33 | (20-45) |
| | *Mode 2* | 216 | (76-432) | 1.47 | (1.28-1.868) | 73 | (49-116) |
| | *Mode 3* | 161 | (83-277) | 1.41 | (1.32-1.542) | 177 | (143-214) |
| **Cluster 2** | *Mode 1* | 5403 | (3381-7557) | 1.41 | (1.29-1.538) | 15 | (11-17) |
| | *Mode 2* | 1560 | (517-3510) | 1.25 | (1.19-1.412) | 24 | (18-45) |
| | *Mode 3* | 439 | (192-1190) | 1.49 | (1.33-1.743) | 73 | (42-104) |
| **Cluster 3** | *Mode 1* | 2038 | (1308-2878) | 1.38 | (1.27-1.524) | 16 | (13-19) |
| | *Mode 2* | 879 | (339-1720) | 1.33 | (1.22-1.655) | 32 | (22-48) |
| | *Mode 3* | 305 | (149-713) | 1.55 | (1.36-1.924) | 88 | (53-128) |
| **Cluster 4** | *Mode 1* | 2920 | (1494-4725) | 1.44 | (1.31-1.597) | 24 | (20-28) |
| | *Mode 2* | 2004 | (690-3407) | 1.32 | (1.22-1.639) | 33 | (29-45) |
| | *Mode 3* | 505 | (250-1173) | 1.44 | (1.3-1.751) | 92 | (52-137) |
| **Cluster 5** | *Mode 1* | 1636 | (829-2512) | 1.45 | (1.3-1.627) | 28 | (23-32) |
| | *Mode 2* | 1046 | (373-1843) | 1.39 | (1.24-1.714) | 41 | (34-64) |
| | *Mode 3* | 321 | (161-646) | 1.48 | (1.33-1.764) | 122 | (69-163) |
| **Cluster 6** | *Mode 1* | 1856 | (920-3032) | 1.42 | (1.29-1.586) | 39 | (31-44) |
| | *Mode 2* | 1613 | (727-2582) | 1.38 | (1.25-1.811) | 53 | (46-65) |
| | *Mode 3* | 370 | (144-946) | 1.39 | (1.26-1.638) | 125 | (71-176) |
| **Cluster 7** | *Mode 1* | 953 | (418-1648) | 1.47 | (1.31-1.679) | 40 | (28-49) |
| | *Mode 2* | 961 | (380-1567) | 1.45 | (1.28-1.866) | 62 | (53-75) |
| | *Mode 3* | 299 | (132-624) | 1.4 | (1.28-1.587) | 152 | (94-195) |
| **Cluster 8** | *Mode 1* | 921 | (470-1339) | 1.52 | (1.34-1.723) | 24 | (18-32) |
| | *Mode 2* | 535 | (167-987) | 1.45 | (1.27-1.815) | 47 | (37-71) |
| | *Mode 3* | 210 | (111-382) | 1.46 | (1.33-1.686) | 140 | (96-175) |
| **Cluster 9** | *Mode 1* | 1088 | (420-2056) | 1.41 | (1.28-1.575) | 50 | (34-60) |
| | *Mode 2* | 1584 | (743-2472) | 1.42 | (1.28-1.757) | 72 | (64-83) |
| | *Mode 3* | 553 | (219-1186) | 1.4 | (1.27-1.62) | 144 | (91-201) |
| **Cluster 10** | *Mode 1* | 423 | (181-827) | 1.53 | (1.35-1.769) | 46 | (27-59) |
| | *Mode 2* | 629 | (253-1021) | 1.48 | (1.29-1.847) | 80 | (65-103) |
| | *Mode 3* | 331 | (167-606) | 1.42 | (1.31-1.57) | 180 | (136-222) |
| **Cluster 11** | *Mode 1* | 528 | (195-1127) | 1.46 | (1.31-1.674) | 52 | (32-69) |
| | *Mode 2* | 1159 | (527-1762) | 1.48 | (1.31-1.78) | 89 | (77-104) |
| | *Mode 3* | 548 | (250-1061) | 1.41 | (1.3-1.573) | 177 | (120-225) |
| **Cluster 12** | *Mode 1* | 817 | (263-1801) | 1.43 | (1.29-1.62) | 56 | (37-74) |
| | *Mode 2* | 1885 | (796-2936) | 1.5 | (1.32-1.771) | 97 | (85-111) |
| | *Mode 3* | 1070 | (454-2050) | 1.43 | (1.31-1.571) | 176 | (127-220) |

**Table 2: Fitted modal parameters per cluster shown in figure 2. GSD represents the geometric standard deviation and Dg the geometric mean diameter of each one of the log-normal modes.**

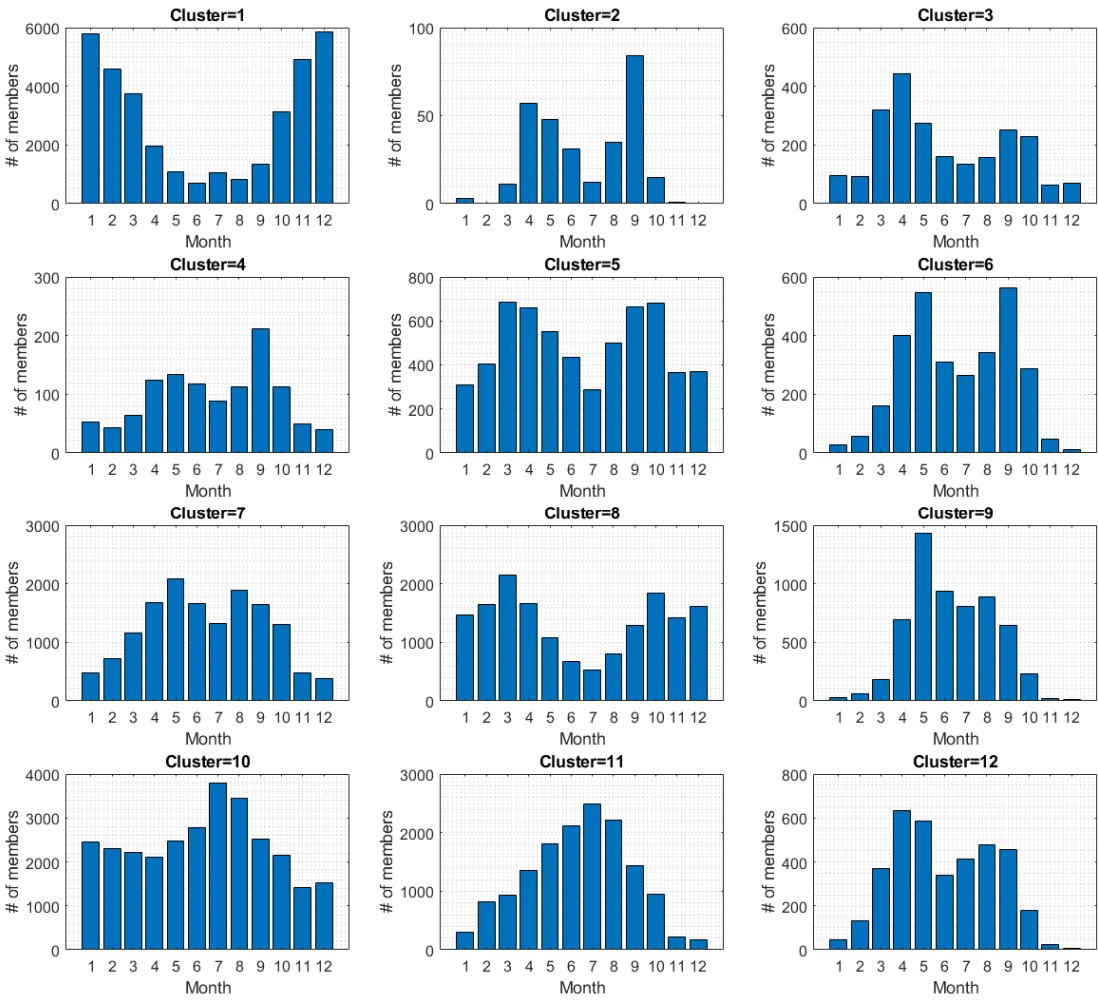

**Figure 4: Seasonal distribution of clusters 1-12. Figure indicate the number of cluster members observed at given month (UTC).**

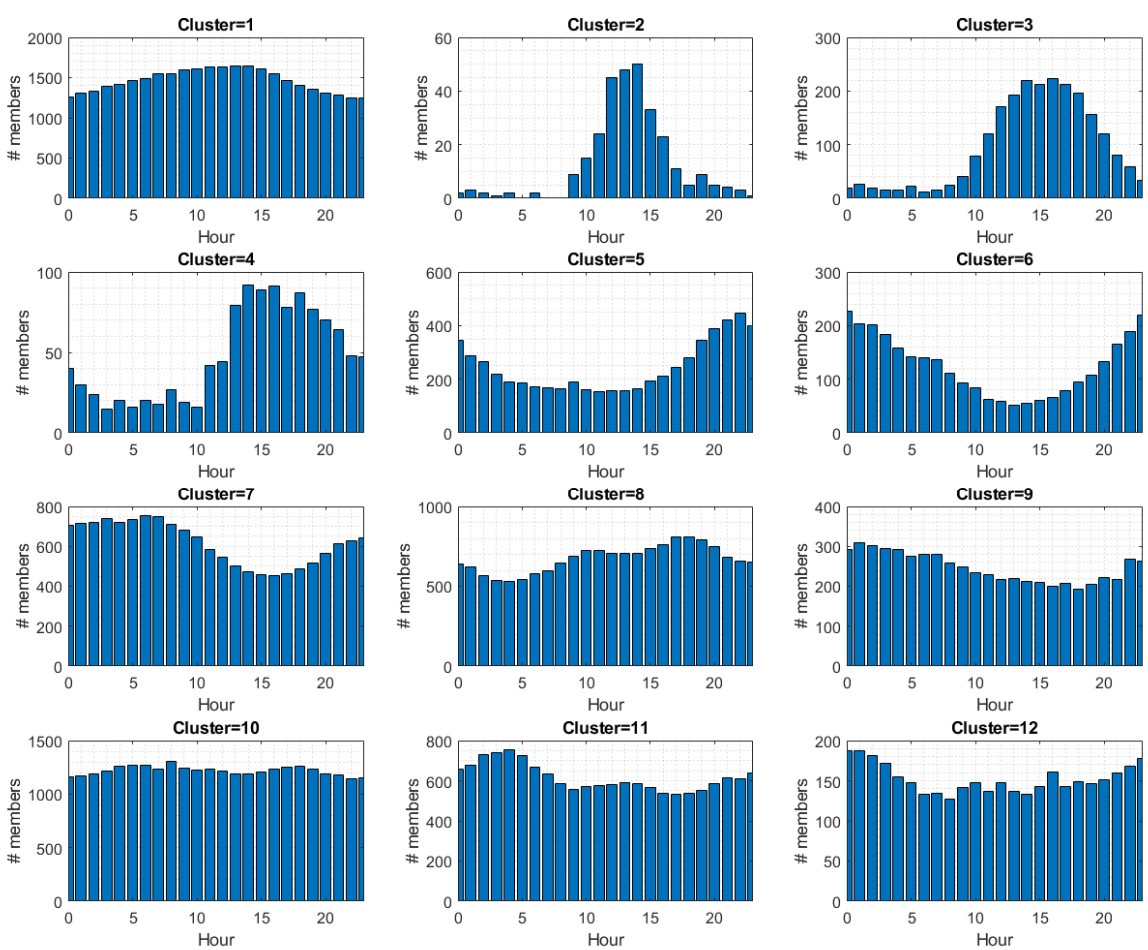

5    **Figure5: Diurnal variability of clusters 1-12. Figure indicate the number of cluster members observed at given time of day (UTC).**

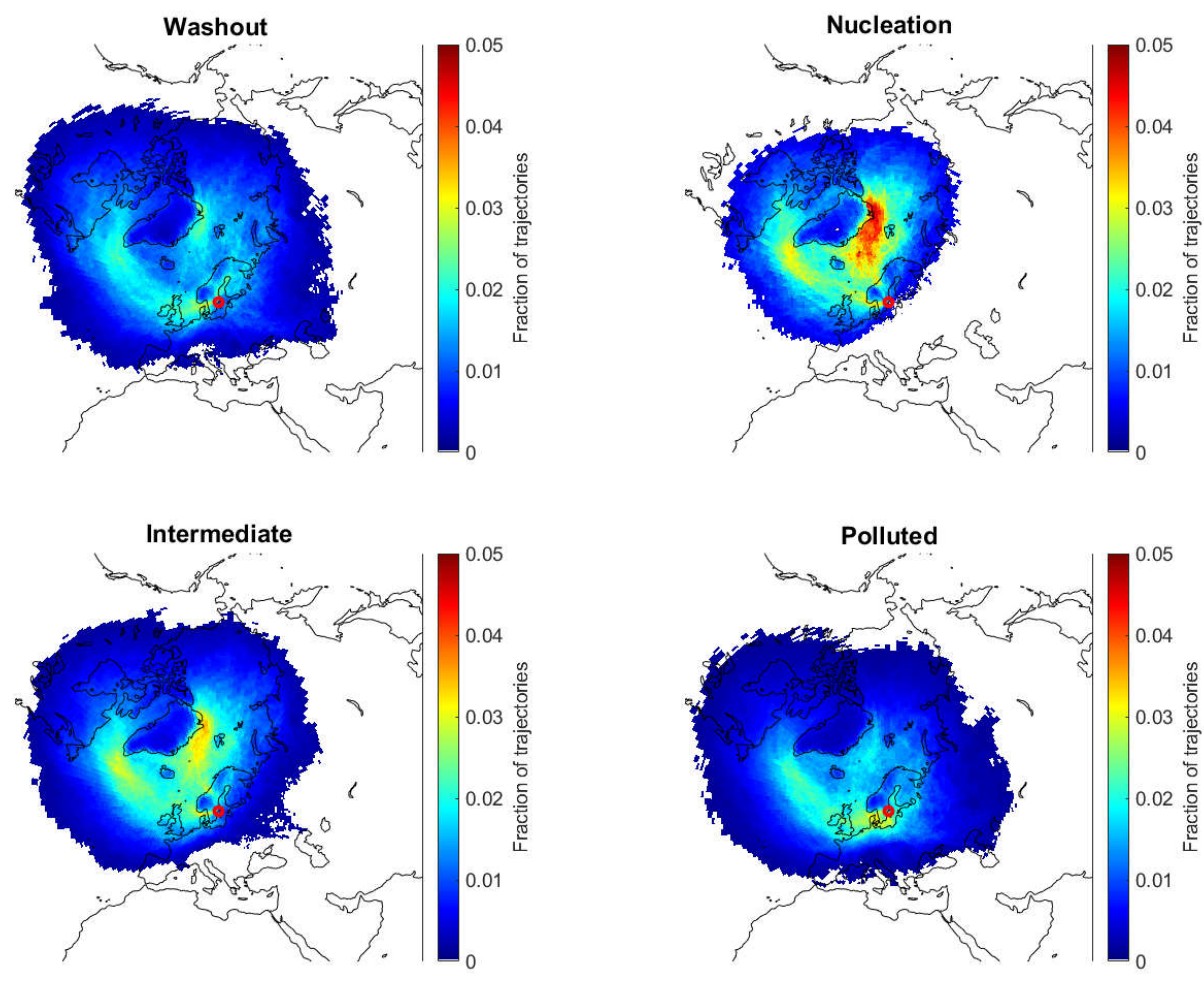

**Figure 6: Dominant source regions of air mass trajectories belonging to cluster groups denoted** *washout* **(Cluster 1),** *nucleation* **(clusters 2 through 6),** *intermediate* **(clusters 7 to 9) and** *polluted* **(clusters 10-12). Aspvreten highlighted by red dot.**

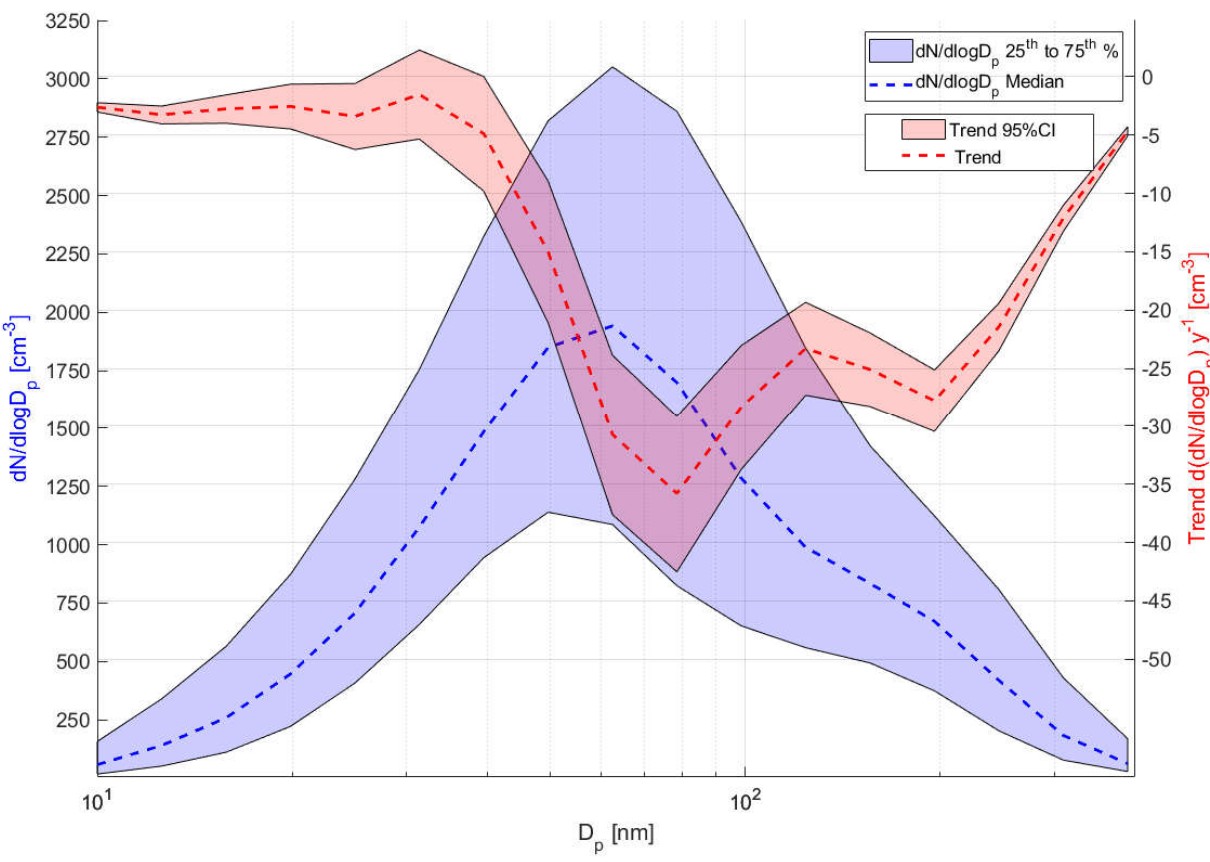

**Figure 7: Calculated annual average trend per size bin for studied period 2000-2017 (right y-axis). The figure also shows median number size distribution and 25-75th percentile interval for reference (left y-axis).**

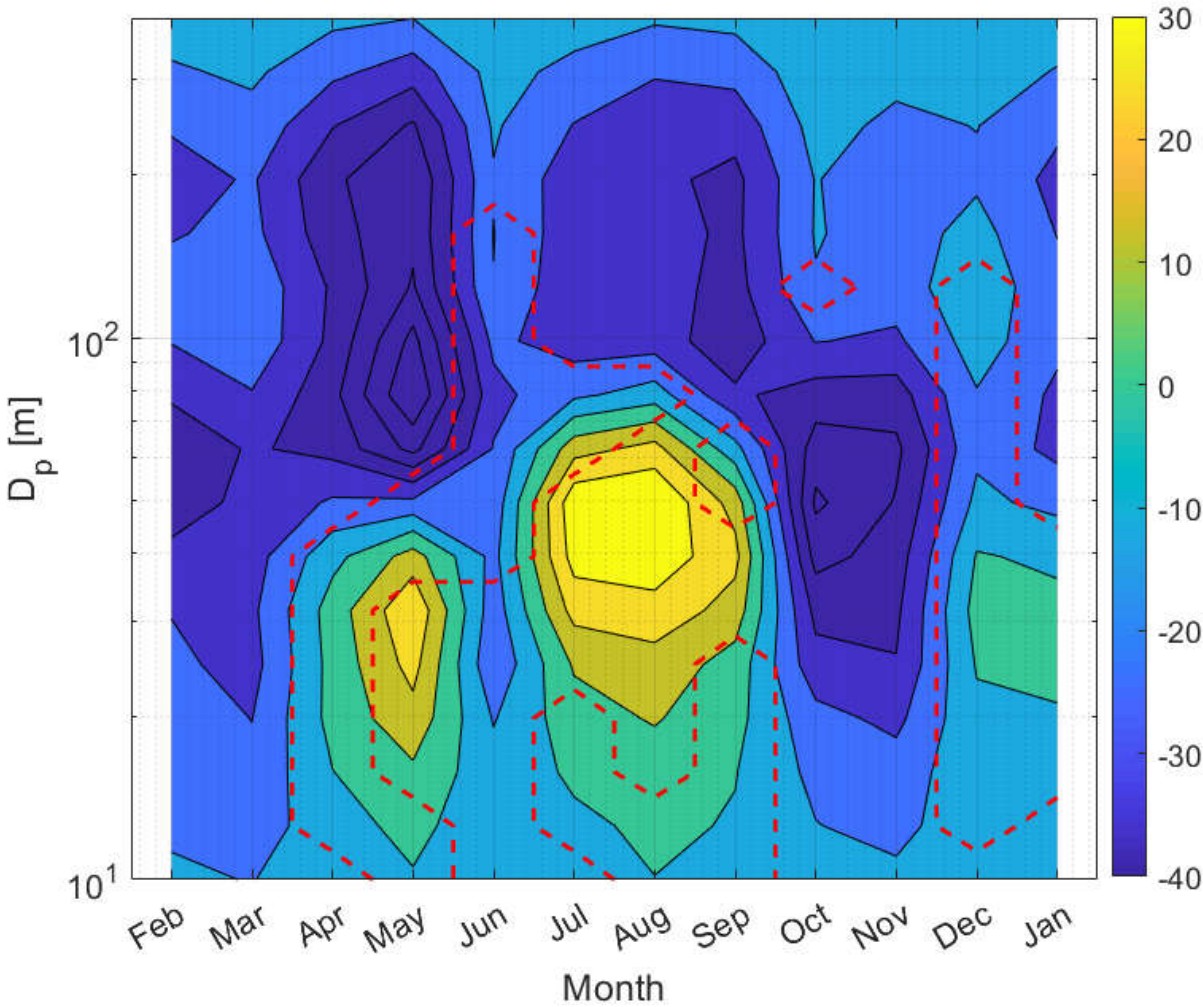

**Figure 8: Contour plot depicting the calculated trends (as Theil-Sen Slopes) for different size classes between 10 and 390nm. Color indicate calculated linear trend for binned particle number concentration at Aspvreten as particle cm$^{-3}$ year$^{-1}$ for the time period 2000-2017. Daily averaged data was used for calculation of Theil-Sens's slope. Areas enclosed by the dashed red line represents pairs of month/size bin where test for significance failed.**

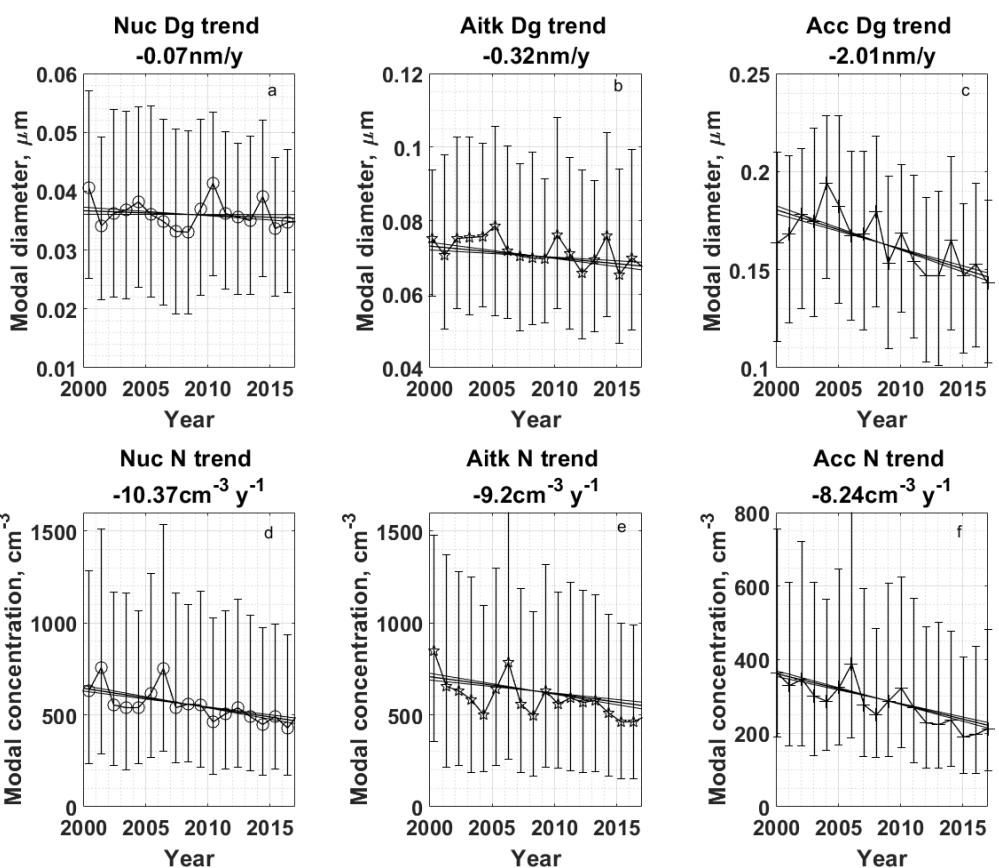

Figure 9: Trends of fitted log normal parameters 2000-2017 of nuclei, Aitken and accumulation modal diameter and modal concentration. Frame a-c shows trends for fitted geometric modal diameter and frame d-f shows trends for fitted modal number concentration. Whiskers indicate 25-75[th] percentile of data. Fitted trend (as Theil-Sen estimator) slope is given in each title. Upper and lower 95% confidence intervals of slopes are indicated in figures. CI of slopes calculated following Hollander and Wolfe (1973).

| | Median Trend ($cm^{-3}y^{-1}$) | Upper CI Trend ($cm^{-3}y^{-1}$) | Lower CI Trend ($cm^{-3}y^{-1}$) | Sigma $\sigma$ |
|---|---|---|---|---|
| $N_{nuc}$ | -10.9 | -12.2 | -8.5 | 0.004 |
| $N_{aitk}$ | -9.2 | -11.4 | -7.1 | 0.0175 |
| $N_{acc}$ | -8.2 | -9.1 | -7.4 | 0.0001 |
| | Median Trend ($nm\ y^{-1}$) | Upper CI Trend ($nm\ y^{-1}$) | Lower CI Trend ($nm\ y^{-1}$) | Sigma $\sigma$ |
| $D_{g,\ nuc}$ | -0.07 | -0.14 | 0.0 | 0.4639 |
| $D_{g,\ Aitk}$ | -0.31 | -0.44 | -0.20 | 0.0069 |
| $D_{g,acc}$ | -2.01 | -2.24 | -1.76 | 0.0011 |

**Table 3: Theil-Sen estimator slope of fitted variables Dg and N and associated upper and lower confidence intervals during time period 2000-2017. Shaded values indicate significance on 0.05 level according to the Seasonal Kendall test (based on 12 months). Calculated σ from seasonal Kendall test is indicated. Trend of Dg for nuclei mode is not significant on the 0.05 significance level. Upper and lower CI calculated following Hollander and Wolfe (1973)**

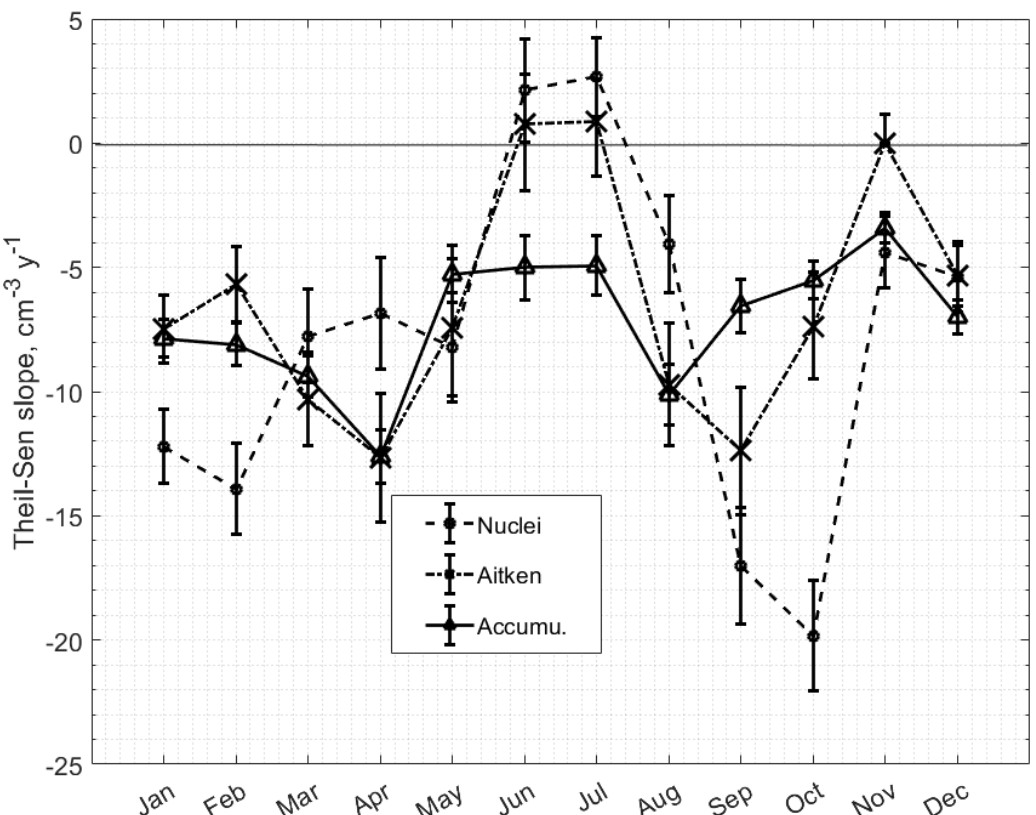

**Figure 10: Theil-Sen estimator slopes for annual trend per month for fitted number concentration of nuclei, Aitken and accumulation modes, January-December 2000-2017. Indicated by error bars are upper and lower confidence intervals of slopes at σ(0.05). All trends tested with Mann-Kendall test and shown significant for σ=0.05 except Aitken mode trend during June-July and November. Upper and lower CI calculated following Hollander and Wolfe (1973)**

| | January | February | March | April | May | June |
|---|---|---|---|---|---|---|
| $N_{nuc}$ | -12.2 (-10.7to-13.7) | -13.9 (-12.1to-15.75) | -7.8 (-5.9to-9.7) | -6.87 (-4.66to-9.12) | -8.2 (-6.0to-10.5) | 2.1 (4.2to0.1) |
| $N_{aitk}$ | -7.5 (-6.1to-8.9) | -5.7 (-4.2to-7.2) | -10.3 (-8.5to-12.3) | -12.7 (-10.1to-15.3) | -7.4 (-4.7to-10.2) | 0.77 (3.5to-1.3) |
| $N_{acc}$ | -7.9 (-7.1to-8.6) | -8.1 (-7.2to-9.0) | -9.40 (-8.6to-10.3) | -12.6 (-11.5to-13.7) | -5.3 (-4.2to-6.4) | -5.00 (-3.7to-6.3) |

| | July | August | September | October | November | December |
|---|---|---|---|---|---|---|
| $N_{nuc}$ | 2.7 (4.3to1.1) | -4.1 (-2.1to-6.0) | -17.0 (-14.7to-19.4) | -20.0 (-17.6to-22.1) | -4.4 (-3.0to-5.8) | -5.4 (-4.0to-6.9) |
| $N_{aitk}$ | 0.9 (3.1to-0.8) | -9.7 (-7.3to-12.2) | -12.4 (-9.8to-15.0) | -7.4 (-5.2to-9.6) | -0.0 (-0.0to-1.2) | -5.4 (-4.1to-6.6) |
| $N_{acc}$ | -4.9 (-3.7to-6.1) | -10.1 (-8.9to-11.4) | -6.6 (-5.5to-7.6) | -5.5 (-4.7to-6.3) | -3.42 (-2.8to-4.0) | -7.0 (-6.3to-7.7) |

**Table 4: Theil-Sen estimator trend slopes ($cm^{-3}y^{-1}$) and upper and lower confidence intervals for monthly inter-annual trends in modal number concentration 2000-2017.**

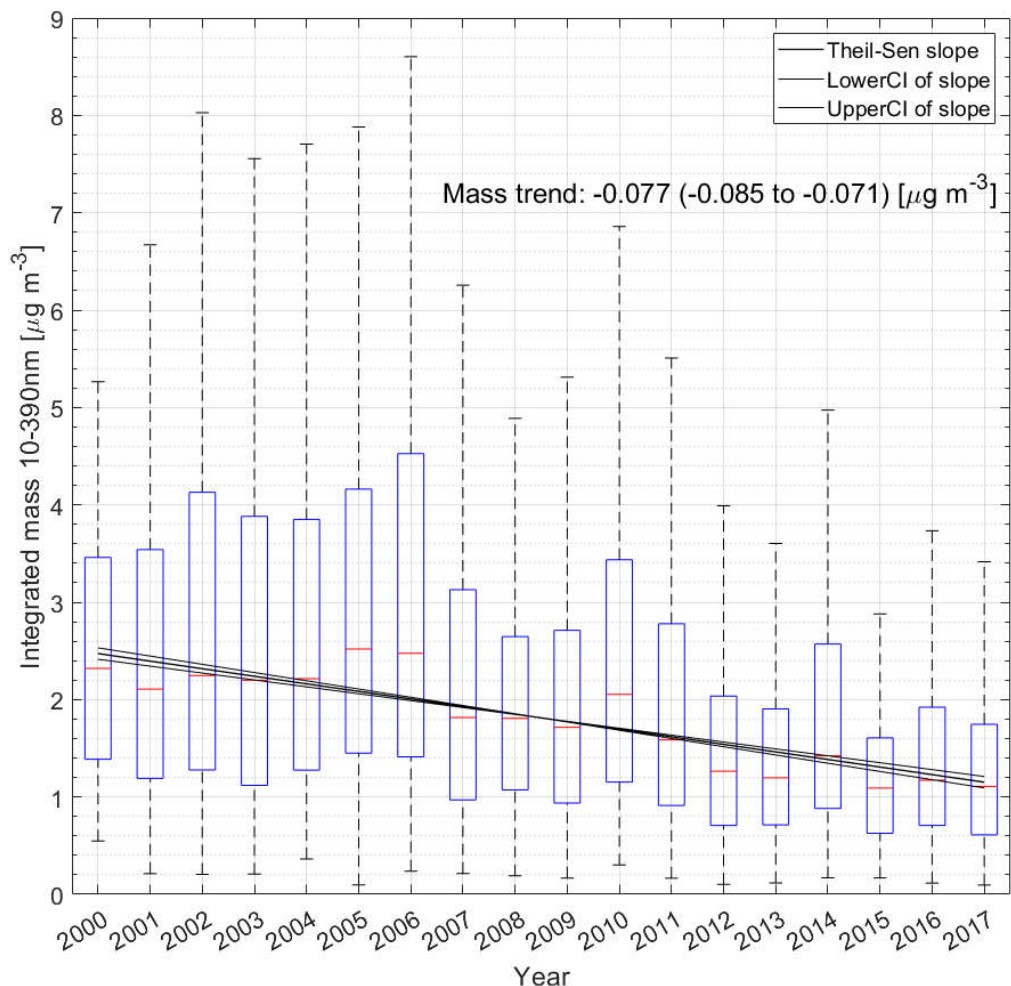

**Figure 11: Box-Whisker plot of daily mean mass concentration 10-390nm assuming density of 1 g cm$^{-3}$ for Aspvreten 2000-2017. Theil-Sen estimator slope and associated 95%CI of slope is indicated in figure by solid black lines. Note that outliers are excluded in the figure for improved clarity. Upper and lower bounds of boxes show 25$^{th}$-75$^{th}$ percentile range, and whiskers 5$^{th}$-95$^{th}$**

10 **percentile ranges of data. Red lines show median.**

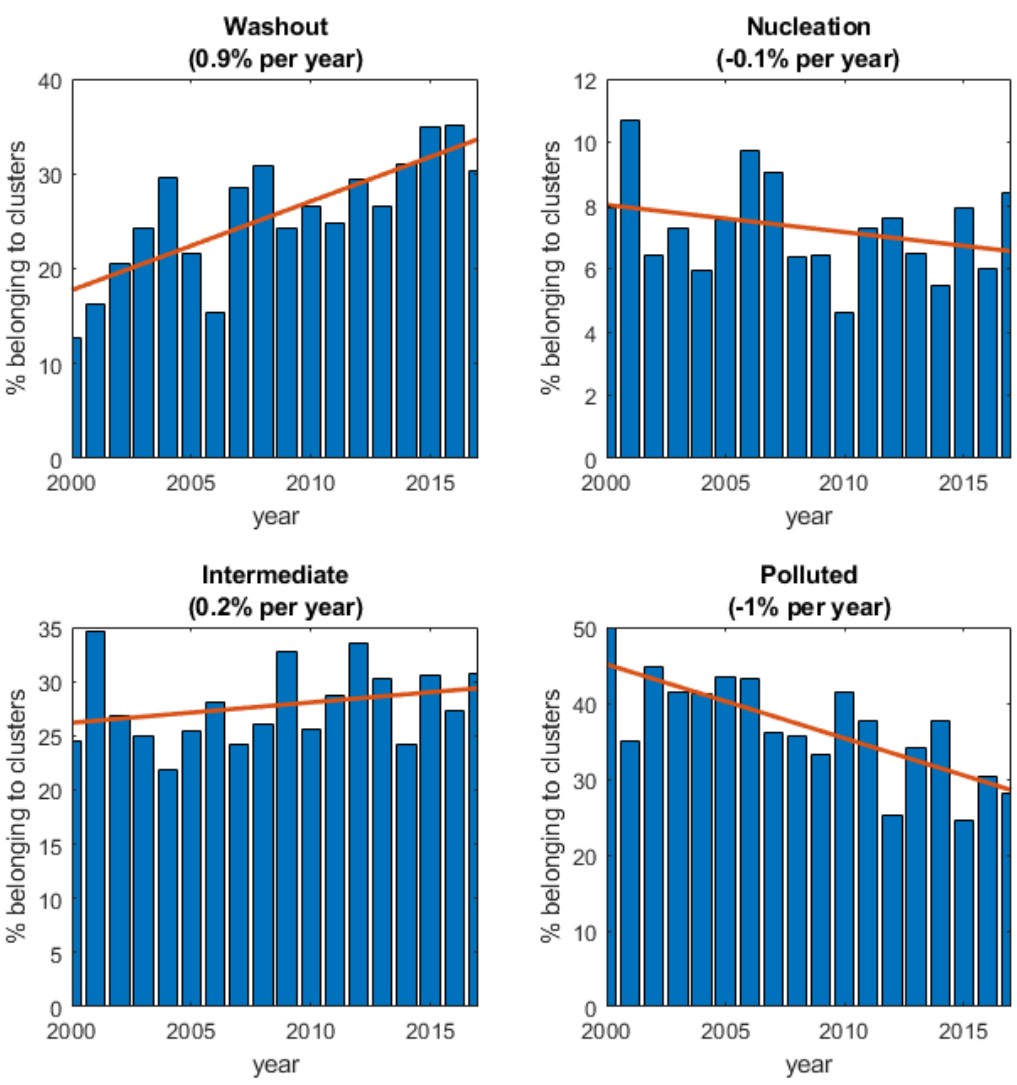

**Figure 12: Inter annual trends of major cluster groups** *washout* **(Cluster 1),** *nucleation* **(clusters 2 through 6),** *intermediate* **(clusters 7 to 9) and** *polluted* **(clusters 10-12). Fitted linear trend and annual decrease is indicated in figure.**

| Updraft (m/s) | Ratio NH$_4$HSO$_4$:ORGANIC | | |
|---|---|---|---|
| | 50:50 | 90:10 | 10:90 |
| **Grand median N$_{drop}$ (cm$^{-3}$)** | | | |
| 0.1m/s | 188(124-264) | 241(169-309) | 90(55-144) |
| 0.5m/s | 409(256-619) | 542(345-790) | 220(138-341) |
| 1m/s | 552(340-847) | 740(463-1096) | 305(192-471) |
| **Linear trends (cm$^{-3}$ year$^{-1}$)** | | | |
| 0.1m/s | -5.0(-5.5to -4.6) | -4.3(-4.8 to -3.9) | -3.5(-3.8 to -3.2) |
| 0.5m/s | -10.3(-11.4 to -9.1) | -10.8(-12.2 to -9.3) | -7.4(-8.1 to -6.7) |
| 1m/s | -13.3(-15.0 to -11.7) | -15.0(-17.0 to -13.0) | -9.3(-10.3 to -8.4) |
| **ΔN/3N (Δτ/τ) 2000-2017 (%)** | | | |
| 0.1m/s | 12.5 | 9.1 | 16.3 |
| 0.5m/s | 12.1 | 10.0 | 14.8 |
| 1m/s | 11.8 | 10.1 | 13.9 |

15  **Table 5: Median (and 25$^{th}$-75$^{th}$ percentile) range (cm$^{-3}$) of calculated CCN's for different updraft and chemical composition scenarios. See text for details. In the calculation of ΔN/3N year 2000 has been used as reference year. Value used is the intercept @ year=2000 of the fitted line. ΔN is calculated as the Theil-Sen slope multiplied with 18 years.**

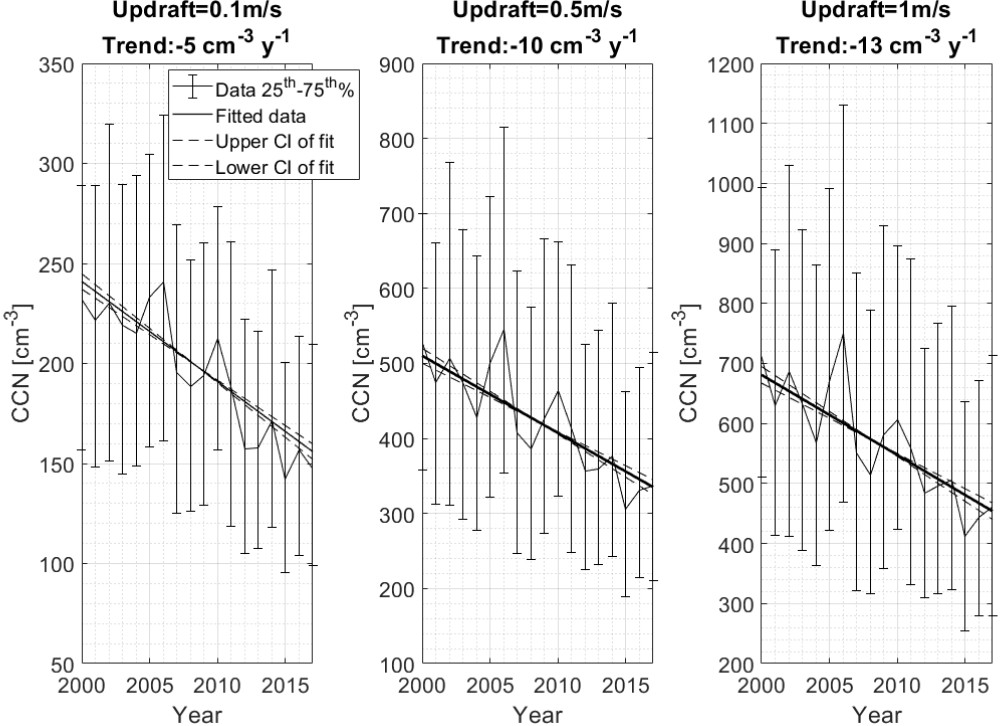

**Figure 13: Trend of calculated CCN for different updrafts assuming a chemical composition ratio NH₄SO₄:Organic of 50:50. Figure shows median calculated CCN per year and corresponding 25th-75th percentile interval. Calculated linear trend as Theil-Sen slope and associated 95% confidence intervals are indicated by solid and dashed lines, respectively. Slope indicated in figure title. See text for further details.**

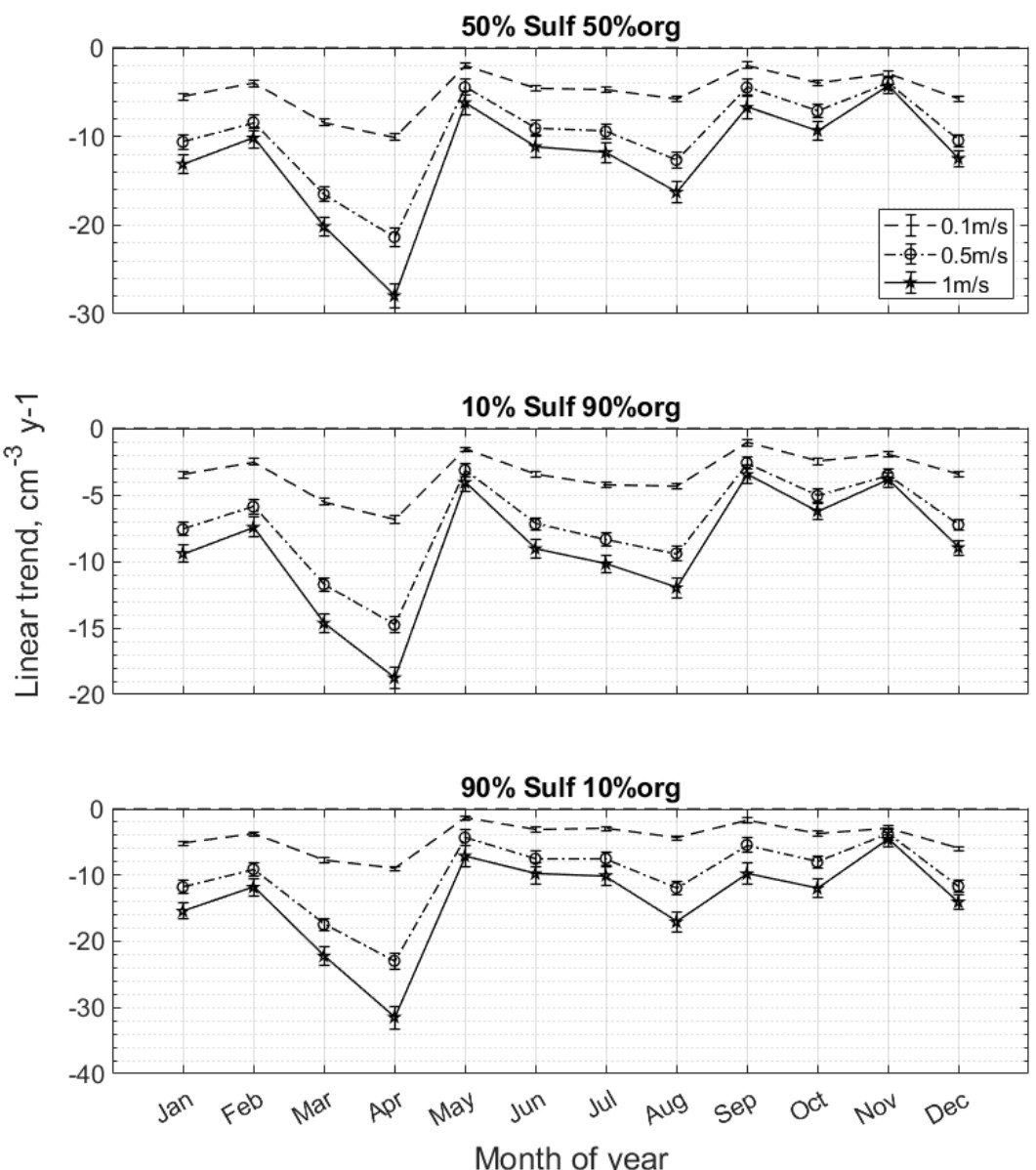

**Figure 14: Calculated Theil-Sen slopes for the 9 different cloud activation scenarios.**

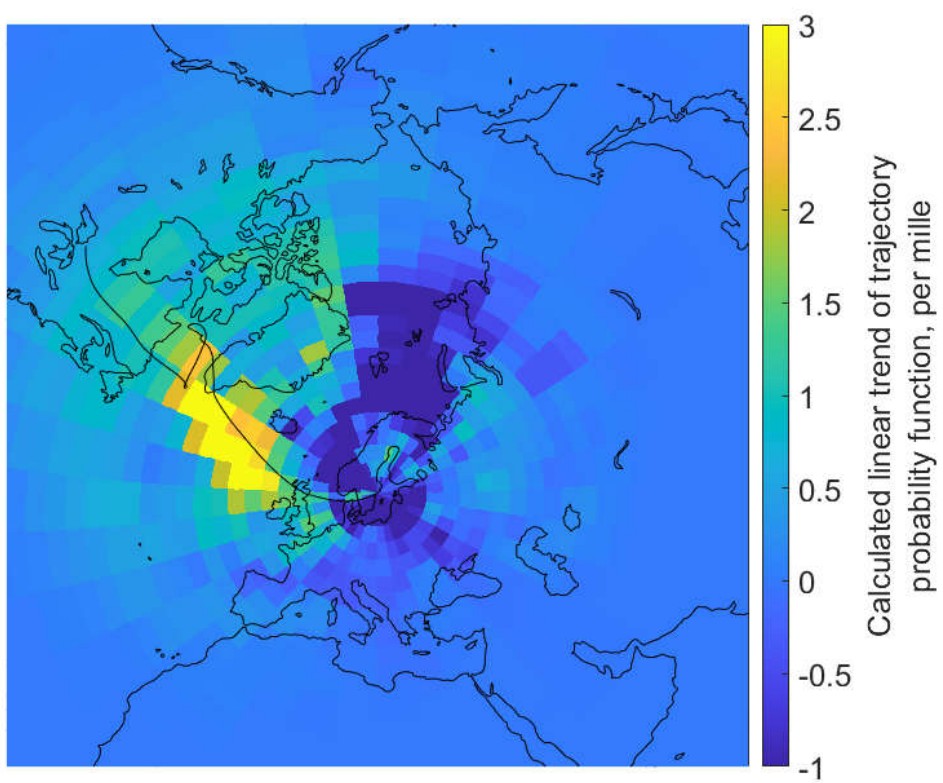

**Figure 15: Trend in probability of transport across different source regions expressed as a linear fit of the trajectory probability function over the 18 year period. Units expressed as *per mille*.**

