# Peer review of "On the seasonal variation of observed size distributions in Northern Europe and their changes with decreasing anthropogenic emissions in Europe: climatology and trend analysis based on 17 years data from Aspyreten, Sweden"

_Atmospheric Chemistry and Physics, 2019_

## Referee Comment (RC1) · Anonymous Referee #1 · 1 Jul 2019

Review – ACPD – Tunved and Ström

**General comments:**

This was a very thorough paper, and generally it looks like a strong study. The authors have obviously put a great deal of work into collecting and analyzing this unique dataset, and in my opinion it is worth publishing, pending that the below concerns are addressed (most of which are fairly minor). Future work could benefit from comparison with other datasets or models.

**Specific comments (most important):**

1. In section 4.5, the authors explore possible variations in transport patterns, but they did not mention how possible changes in wet deposition and cloud processing might also influence the aerosol concentrations, sources, and distributions. I understand that cloudiness and precipitation may concurrently be changing over the Arctic (e.g., see Morrison et al. (2018), Bintanja and Selten (2014)), although I am not sure about at or upwind of the Aspvreten site in particular. The authors might consider adding some information on this in the introduction, and discussing how changes in cloudiness and precipitation might impact the interpretation of their results.

2. I suggest the authors re-visit the text in the following places to make sure they are not over-emphasizing the certainty in how the results should be interpreted.

   - 2.3: Can the authors please address whether the clustering assumptions have the potential to have a major impact on the findings, and if so, whether any sensitivity analyses were conducted to make sure that the findings are robust?
   - P. 8, l. 32: "Absence of small particles signifies the lack of photochemistry." There can't be any other explanation?
   - Figure 5: Is it possible that strong differences in the amount of light at different hours, depending on season, would skew or bias the results presented in Figure 5? Are the results similar when separated be season?
   - The cluster history discussion in section 3.2: The authors make reasonable assumptions about the important atmospheric processes each cluster was exposed to, based on theoretical suppositions about the factors that drive aerosol size distributions. However, up to this point, the data presented are entirely based on size distributions alone with no other supporting data. I would suggest making it clearer in this section that the processes causing the aerosol size distributions are merely hypothesized (unless the authors can present or discuss other datasets supporting these suppositions further).
   - P12 l.1: "The presence of the two modes is suggestive of two different processes acting independent on the size distribution." This statement seems to imply to me that there are not other interpretations as well. Is that what was intended here, and if so, why?
   - Figure 14. I am wondering, and presumably other readers may also wonder, what the implications are for the authors assuming cloud droplet activation in warm clouds at such high latitudes, where ice-containing clouds are very common,

particularly during the winter. Can the authors comment on whether their results might be less applicable or have greater uncertainty during the winter periods? Can the authors provide any information on how common liquid-only clouds are in this region during the different periods of the year?

- Section 4.5 This part of the analysis assumes that the HYSLPIT back trajectories are correct, and equally correct, at each location and time. I am skeptical about whether this is a good assumption, as the Arctic is a generally poorly validated region. As such, it is unclear whether the results are meaningful, although the fact that signals are small might still be worth showing. I recommend either removing this section, moving it to a supplement, or adding substantially more discussion of the uncertainties.

- P. 17, l. 9: "The data has been analyzed from several different aspects in order to shed light on the degree, when and where changes in emissions (natural or anthropogenic) may have had an impact on the observed aerosol size distribution." The authors have only back trajectories to assess the sources of the aerosols in this study. To not oversell the study, I suggest removing the "(natural or anthropogenic)" part of this sentence.

- P. 13, l. 28: "This means a reduction of mass of roughly 52 % during the studied period." I recommend re-wording this sentence to reflect the uncertainties in this statement. One suggestion: "Based on this calculation, we estimate that mass was reduced by roughly 52 % during the studied period"

- P. 18, l. 1-8: How would changes in precipitation impact these findings, if at all? Please briefly discuss.

3. There were several places where the methods were insufficiently described. The authors should please provide the following information:

- Methods: Readers are pointed to outside references for basic information about the sampling. To avoid readers having to go look up a different paper to interpret this one, it would be helpful to at least mention in the text whether these are ground-based data, at what altitude the data were collected, whether there were any potential upwind sources of contamination, and if so, what was done to avoid sampling these sources.

- The trends the authors describe are based on a single instrument, if I understand the methods correctly. Can the authors please mention whether any measures were taken to rule out the possibility that the observed long-term trends are caused by errors in the instrument calibration? (Presumably the instrument was independently calibrated, but it may be useful to compare the data to other long-term satellite or ground-based remote sensing data in the region (e.g., MODIS or AERONET data) as well?)

- Tables 1-2: Please define GSD and Dg in the caption and in the text as well. Also, how is modal diameter and concentration defined, specifically?

- Fig. 7: The caption is confusing because there is only one panel in the figure; please rephrase. The figure is also confusing because the axis on the right labelled "Trend d(dN/dlogDp)" is in reverse order (5 to -50, bottom-to-top) compared to the axis on the left labelled "dN/dlogDp", which goes from 0 to 3250, bottom-totop). It would help readers if the plot were standardized so that both axes were going in the same direction. Please indicate in the text how the confidence intervals were calculated.

- Figure 8. Please indicate how the confidence interval for the trends were calculated, and which of the plotted trends are significantly different from zero. To address the significance, the authors might consider obtaining a confidence interval around the slope for each subset of the data, similar to what was done in Fig. 7 (preferably a bootstrapped confidence interval, as that would be valid even if the data don't meet all the assumptions of a normal linear regression analysis).
- P. 12, l. 33 and Table 3 – It may be helpful to describe (or at minimum, reference) the seasonal Kendall test used here, for those readers who are not familiar with it, and to explain why this test was chosen.
- Figure 9. Please explain in the text how the confidence intervals for the slope were calculated. Please indicate in the caption what the whiskers indicate (range of the data? SD? Etc.)
- Figure 10. Please indicate in the caption what the whiskers indicate.
- Figure 11: Please indicate in the caption what statistics are being presented in the boxes and whiskers. For example, that median, upper and lower quartiles, and some kind of confidence interval?

**Specific comments (more minor):**

In the abstract, the authors might consider more clearly stating the extent to which this work is new and different from previous published work on this dataset, and more general reasons why the work is important.

P.3 l. 23: "Clearly there is a lack of long-term aerosol microphysical data from which trend analysis has been reported or even can be reported because of too short measurement periods or interrupted data sets." To clarify, can the authors please state whether there are similar observations in other locations, and/or what specific region the above statement is referring to?

Fig. 6: please add the location of the sampling site to this map.

P. 12, l. 2: "As sulfur is the particle precursor that has decreased the most …" please cite what information this statement is based on.

P. 16, l.2: "However, our analysis above also emphasize that the apparent trends are not the same over the whole year." Apparent trends in what?

p. 17, l. 15: "The general trend found in this study is well in agreement with the findings presented by Asmi et al. (2013)…." Please mention here where the Asmi et al. study took place.

**References**

Bintanja, R. and Selten, F. M.: Future increases in Arctic precipitation linked to local evaporation and sea-ice retreat, Nature, 509(7501), 479–482, doi:10.1038/nature13259, 2014.

Morrison, A. L., Kay, J. E., Frey, W. R., Chepfer, H. and Guzman, R.: Cloud Response to Arctic Sea Ice Loss and Implications for Future Feedback in the CESM1 Climate Model, Journal of Geophysical Research: Atmospheres, 124(2), 1003–1020, doi:10.1029/2018JD029142, 2018.

---

## Referee Comment (RC2) · Anonymous Referee #2 · 11 Jul 2019

Review of "On the seasonal variation of observed size distributions in Northern Europe and their changes with decreasing anthropogenic emissions in Europe: climatology and trend analysis based on 17 years data from Aspvreten, Sweden", by Peter Tunved and Johan Stroem. (submitted to Atmos. Chem. Phys.).

This manuscript presents an analysis of 17 years of near-surface aerosol particle size distribution measurements at the Aspreveten regional background monitoring site in

[Figure]

Central/Southern Sweden. The topic of the research is of particular interest to understand how the tropoapheric aerosol layer has responded to reductions in emissions that have occurred in the 1980s and 1990s, re: subsequent climate influences from aerosol-radiation-interaction and aerosol-cloud-interaction radiative effects.

The paper represents an interesting and novel study of the 17 years of size distribution measurements, with trends applied on different types of size distribution, identified via cluster analysis. The approach may represent an important advance, the technique complimenting other studies size-resolved aerosol trend analysis studies, stratifying into size-distribution-types then enabling to assess process-linked changes, with then potential to explore for any signals of tropospheric aerosol layer response to reduced emissions in recent decadaes.

However, the Abstract and Introduction in the current version of the manuscript require substantial improvement and the initial summary statistics will be confusing to some readers as presently worded. The authors need to improve the explanation of these initial overview comparison plots to put the Asprevreten site, and the occurrence of new particle formation events, into better context in comparison to other size distribution monitoring sites in Scandinavia: Hyytiala, Pallas, Varrio and Vavihill.

It remains surprising to me that the seasonal average size distributions in Figure 2 seemed to suggest only two sub-micron modes – an Aitken mode at about 50-80nm diameter and an accumulation mode at about 200nm diameter, and yet the statistics from log-normal fits to the hourly measurements in Table 1 identify three modes – with a distinct nucleation mode at 30-40nm, in addition to an Aitken mode at 60-70nm.

The daily-mean size distribution overview plot in Figure 1 shows a clear seasonal variation in the aerosol measured at Aspreveten, total number concentrations increasing from 1000 per cc in winter to about 2000 per cc in summer, with Aitken mode peak at Dp=50nm in winter, increasing to 100nm in summer. And at least from these daily-mean plots in Figures 1 and 2, there is no obvious sign of a nucleation mode, whereas

in Table 2 the nucleation mode has approximately equal particle concentrations to the Aitken mode.

The manuscript needs to explain the reason for this apparent discrepancy, and also put the Aspreveten site analysed here into better context with the other Scandinavian sites.

Strong diurnal variation occurs during new particle formation (NPF) events (e.g. as in Figure 2 of Kulmala et al., 2004), from initial NPF early-morning and subsequent growth through the morning and early-afternoon, the air measured through the day then having different times from growth then manifesting as the so-called banana plots in dN/dlogr vs time within the air sampled at the monitoring site. And it is likely the case then that these NPF variations will be less apparent within daily-mean or daily-integrated aerosol measurements. However, the difference in these statistics may be underlining the importance of applying the trends analysis on the cluster-analysis-stratified data, compared to what is apparent from more daily-mean data, which will average out much of these important variations.

Is it the case that Aspreveten has weaker or less frequent nucleation days than Hyytiala and other near-forest sites, perhaps due to much less biogenic VOC emissions from forests, and subsequent influence on new particle formation (e.g. Metzger et al., 2010)? Or is it simply that the nucleation events are occurring, but daily-means smooth out the strong diurnal variation in NPF events?

If it is the latter, there may be important implications for the way global modelers design model experiments, I mean in terms of diagnostics to retain these important variations. Often modellers are ambitious in attempting to be comprehensive, to compare measurements to a large number of datasets, and likely consider a compromise necessary in choices of diagnostic information to include in the model experiments. However, in the case of new particle formation it may be a pre-requisite to ensure information on the variability on an hourly timescale is retained (perhaps with some approach to store the

mean of the square-of-a-metric as well as the mean of a metric to enable the variability to be reconstructed, with some optimal sampling and/or re-initialisation sequence).

I am not suggesting the authors make this specific point, but, if the statistical variation is indeed the reason for the apparent discrepancy, to consider adding a more general note, either in the discussion or conclusion-bullet-points about any re: how modellers can efficiently/effectively retain the diurnal variation information in their simulations. For example a statement along the lines of the hourly-mean and daily-mean statistics here underline the importance to consider how best to design diagnostics in models to retain information on new particle formation within co-ordinated composition-climate model experiments.

My other major concern is re: section 4.4 where the CALM model from Tunved et al. (2010) is applied, and some additional calculations are made to estimate the potential cloud albedo change of the trend period. I could not follow the calculations on page 15 of the manuscript, and do not see a reference for the application of the CALM model for this application. I suggest the authors consider whether it might be best to restrict the scope of the article to describing the aerosol changes.

It is my opinion that, with some additional revisions (which I classifiy here as major, but may actually be relatively straightforward to make) some important conclusions can be identified from combining the cluster analysis and the 17-year trend analysis. My recommendation is the authors restrict the scope of this paper to the aerosol changes and consider whether co-operation from others might enable the approach to potentially be applied also on other sites with long-term trends (e.g. those in Asmi et al., 2013)? In that scenario, the article could well stand independently from section 4.4, the analysis potentially benefitting also from a more focussed scope on this single site, and also identifying a potential future article.

To summarise, my review is to recommend major revisions, including re-drawing Figure 1 adding in an extra sub-panel (Figure 1b, see point 30 below) to present the variability

within the daily-means – perhaps simply the ratio of the standard deviation to the mean would provide, via the coefficient of variation, an indication of which parts of the particle size range have substantial variability, likely highlighting days where new particle formation and growth is occurring upwind of the site. I think this could be a good way to provide overview analysis of the seasonal occurence of these events, and link then to explain the discrepancy with the hourly-mean statistics in Table 1.

I have restricted the scope of this review mostly to the Abstract, Introduction and initial Figures – the above issues seeming to me to require a further substantial revision to the manuscript, but can confirm I am willing to review the revised manuscript once the specific revisions I recommend below have been made.

Specific revisions ——————

1) Abstract, line 9 – add "at a rural background site in Northern Europe" (or similar) after "trends", and corrected grammatical error "investigate" –> "investigated".

2) Abstract, line 13 – delete "has been" and be more specific when you say "during last decades" – maybe it's just a case of replacing with "during the 17-year trend period"? Also, I think one of the key things in your findings is that your analysis identifies that particle size distribution has shifted as well as the particle number. The previous N20 and N100 trends may also show this to some extent, but the size-resolved cluster analysis at this single site here may be better able 17 to demonstrate this? So you could maybe strengthen the point about particle size by "We show that, not only have particle number concentrations decreased, but also particle size has shifted, with potential implications for aerosol-climate influence?" You could also add something to the conclusions re: future work to further investigate this at other sites?

3) Abstract, lines 14-15 – the 2 sentences here should be joined into a single sentence, so that then the two sentences following on from the previous point are first one re: changes in particle number, and then the 2nd one re: the shift in particle size? The wording just needs to be improved, if possible.

[Figure]

4) Abstract, lines 17-18 – this setence again should be strengthened to make the point that the shift in size affects mass, but add also re: the importance then for radiative effects.

5) Abstract, lines 19-22 – rather than stating "a rather complex picture emerges" these sentences here need to summarise specifically the seasons where the main decreasing trend is seen – then point out which seasons or modes show the increases (is it just, Aitken mode and summer?)

6) Abstract, lines 22-24 – all previous text has referred specifically to the measurements at Aspreveten, but this sentence is (I think) referring to the trajectory model analysis you've carried out? If that's correct then rather than "data analysis", instead give additional/different words to explain that it is from simulations with the trajectory that this finding is demonstrated? Or if it is simply your interpretation of this, then state "We interpret this as.. " or similar. Also – re: the "receptor" – here you mean the actual Aspreveten site is a receptor – maybe better to explain this as "receptor site" or "upwind of the receptor site"?

7) Abstract, lines 24-26 – is "an adiabatic cloud parcel model" the right description for the analysis tool applied for section 4.4? The CALM model described in Tunved et al. (2010) seems to be predicting the evolution of the size-resolved aerosol with some influence from clouds. The simulations can predict the CCN considering air-parcel trajectories, but it is, to my understanding, not really a cloud parcel model. Please re-word this sentence, and the opening paragraph to section 4.4 accordingly.

8) Abstract, lines 26-27 – as per my main comments above, I am recommending the authors restrict the scope of this article to the aerosol changes, in which case this additional finding re: the 10-12% cloud albedo change may not need to be included.

9) Introduction, page 2, lines 2-4 – the choice of "Turbidity" in this opening sentence was unusual and is, in my opinion, a good alternative word to describe the optical thickness of the atmosphere. The word was often used in the articles in the 1960s

and 1970s, for example re: the dimming effect from the volcanic aerosol haze after the 1963 Agung eruption (e.g. Volz et al., 1970), but these days is seldom used in journal articles. The original 1908 article by Gustav Mie used this term to describe the aerosol-radiation interaction effect (see e.g. Lilienfeld et al., 1991) and it is, in my opinion, good terminology. However, in the 2nd sentence, with the Mie scattering from aerosol, and the associated radiative effects from aerosol changes these days a core requirement within all climate model integrations, suggest to refer to either a review article and/or relevant text book (e.g. Seinfeld and Pandis, 1998) when introducing these basic effects for explaining the solar dimming and the relationship to the size-resolved aerosol. Also, in addition to explaining a correspondence with aerosol volume in the accumulation mode, suggest to add these days the understanding of the origin of such particles being from sizes as small as a few nm, and to understand the importance of new particle formation (e.g. cite Kulmala et al., 2004 for the measurements that have demonstrated this).

10) Introduction, page 2, line 9 – Please follow the recommended way to cite IPCC climate assessment reports, via citing the relevant chapter, e.g. Mhyre et al. (2013) for the radiative forcing chapter.

11) Introduction, page 2, line 10 – suggest to change "indicate that dimming" to "show that overall solar dimming"

12) Introduction, page 2, lines 11-13 – re-word this sentence to more clearly explain the brightening trend – it is not clear what you mean by "effect of economic growth" – I think you mean economic growth in East and South Asia, with the 2nd part of that sentence on air quality legislation being re: emissions reductions Europe and North America. The overall brightening effect suggests the aerosol decreasing overall, and you need to clarify this sentence to explain this better, perhaps citing other articles such as Ohmura (2009). Although the next paragraph indicates the regional changes, this initial sentence needs to be clearer.

13) Introduction, page 2, lines 15-16 – add "vapour" after "sulfuric acid" in both instances here, so be clearer re: the subsequent condensation onto particles and driver of new particle formation. Suggest also replace "sulfate aerosol mass" with "sulfate aerosol particles" or "sulfate aerosol particle mass" so it's clear you mean the aerosol particle phase. Suggest to change "condensation onto already existing particles" with "condensation, which grows existing particles".

14) Introduction – page 2, line 19 – re-word "with the exception perhaps of the Indian sub-continent". Although there are uncertainties you can be more certain about the different trends in emissions and suggest to describe the regions more regionally such as South Asia, citing a recent paper on this, and perhaps draw a distinction between aerosol precursor emissions and observed changes re: other emissions such as carbonaceous aerosol and influences from other precursor emissions or oxidation processes.

15) Introduction – page 2, line 23 – the term AOD refers to integrated aerosol extinction, and hence just the "aerosol optical depth" not "atmospheric optical depth". Please correct this.

16) Introduction – page 2, line 25 – this seems too detailed information on North America, and more detail on any differences in trends within Europe would be better here – does the literature contrast trends in different parts of Europe (e.g. contrast Northern Europe to those observed in Central Europe and in Southern Europe?).

17) Introduction – page 2, line 29 – suggest to delete the trends in Arctic and Antarctic to give room for the above suggested discussion re: any literature on different aerosol trends within Europe.

18) Introduction – page 2, line 30-34 – reduce this para on Turnock et al. to focus on the main findings, again add some additional discussion re: aerosol changes in different regions of Europe.

19) Introduction – page 3, lines 10-12 – reword "However, due to claimed lack of data" – to explain instead as sites with more than 10 years of data for the trend analysis. Also there were 5 sites in that study, not 4 – Melpitz in Germany was also included in the analysis.

20) Introduction – page 3, lines 23 – reword "Clearly there is lack of long-term aerosol microphysical data" to be more positive. These days, following the programs to establish additional sites to monitor aerosol size distribution, there are several additional sites that have 10 years of data, for example the EUSAAR sites analysed in Asmi et al. (2011), with SMPS/DMPS aerosol measurements since the time of the EUCAARI EU FP6 integrated program (Kulmala et al., 2009).

21) Introduction – page 3, lines 27-30 – reword this sentence maybe to be clearer you mean these monitoring sites are providing valuable new understanding of aerosols, and in particular trends in Northern Europe. Maybe the start of that sentence add "in Scandinavia" (rather than later) or "in Northern Europe", e.g. re: where 4 of the 5 sites in Asmi et al. (2013) analysis are located.

22) Introduction – page 4, line 16 – "The explanation for this" – cite a paper that has already shown or explained this effect – perhaps cite Kerminen and Kulmala (2003) or other paper re: the role of the condensation sink and new particle formation?

23) Introduction – page 4, line 19 – "Based on the literature referenced above there is a notion" Re-word this to be clearer citing relevant paper etc. I think you can be clearer to state this is well-established to be the case. There is a general question of the "aerosol response" to the reduced SO2 emissions in different parts of Europe and perhaps this part of the Introduction you could bring the earlier literature review round to bigger-picture question and then explain the specific analysis this paper presents.

24) Methods – page 5, lines 19-20 – order the 2003, 2004, 2005 papers chronologically here.

[Figure]

25) Results – page 8, line 32 – explain a bit more re: the link between the lack of photochemistry and the absence of small particles – you could maybe state the oxidation of SO2 specifically here and the short atmospheric residence time of the sulphuric acid vapour, or so.

26) Summary and conclusions – page 19, lines 1-4 Replace "increasing on expense of" with "with a reduction in" and replace "vanished during the period 2000-2017" with "reduced to xx% at the end of the trend period" or similar.

27) Summary and conclusions – page 19, lines 23-26 The text for this bullet begins stating no evidence for increased nucleation was found. But then the last sentence says there was more frequent nucleation upstream of the site. Please clarify – as per earlier comments, I think the finding of increased nucleation is important one and perhaps the revised paper could focus more on identifying this and adding re: future work to understanding the overall aerosol reponse in other sites and regions of Europe?

28) Summary and conclusions – page 19, lines 29-32 – and page 20, lines 1-5 As per my main comments, I wonder whether best to restrict the analysis to the aerosol trends rather than the activation and cloud albedo calculations? These final 2 bullet points (particularly the last one) do not add substantial findings – suggest better to focus on the main conclusions re: the aerosol trends.

29) References – the References section seems to have reverted back to a different font. Please remedy this to match the required style for ACP.

30) Figure 1, page 24 – as per my main comments, I was trying to understand why the daily-mean size distributions seem (from this initial Figure) to suggest only one mode across the nucleation and Aitken, whereas the fits to the hourly measurements in Table 1 then reveal the presence of the distinct nucleation mode. I'm suggesting here whether showing as an extra panel b) in this Figure you could show the relative variability within the 24 hours of data (for each size bin) – maybe plot the ratio of the standard deviation to the mean – i.e. the coefficient of variation – to show which parts

of the size range then have most variability – perhaps this will then identify this variation and reason why the nucleation and Aitken modes in Table 1 are only apparent in the hourly data?

31) Figure 2, page 25 – the presence of the accumulation mode in these seasonal size distribution pdfs is apparent on this linear y axis, but it would be clearer if the plot showed these size distributions on a log-y axis. This also relates to the trends in particle number, with a number concentration change of 10 per cc being more significant for the accumulation mode than for the Aitken or nucleation modes. Suggest to re-plot this Figure with the dN/dlogr axis showing then 10 to 10,000 per cc or so, to be able to see the variations in number in the accumulation mode, as well as those in the Aitken mode.

32) Table 1, page 26 – the median and inter-quartile range for the nuclei and Aitken modes are similar, from these hourly size distribution log-normal fits, but the presence of the 2 modes is not apparent in the daily-mean size distributions in Figures 1 and 2. See main comments, and points 30 and 31.

References ————-

Asmi, A., Wiedensohler, A. Laj, P. et al. (2011) Number size distributions and seasonality of submicron particles in Europe 2008–2009, Atmos. Chem. Phys., vol. 11, pp. 5505–5538.

Asmi, A., Collaud Coen, M. Ogren, J. A. et al. (2013) Aerosol decadal trends – Part 2: In-situ aerosol particle number concentrations at GAW and ACTRIS stations Atmos. Chem. Phys., vol. 13, pp. 895–916.

Kerminen V.-M. and Kulmala, M. (2002) J. Aerosol. Sci., vol. 33, pp. 609–622.

Kulmala, M., Vehkamaki, H. Petajaa, T. et al. (2004) Formation and growth rates of ultrafine atmospheric particles: a review of observations Journal of Aerosol Science, vol. 35, pp. 143–176.

Kulmala, M., Asmi, A., Lappalainen, H. K. et al. (2009) Introduction: European Integrated Project on Aerosol Cloud Climate and Air Quality interactions (EUCAARI) - integrating aerosol research from nano to global scales, Atmos. Chem. Phys., vol. 9, 2825–2841.

Lilienfeld, P. (1991) Gustav Mie: the person Applied Optics, vol. 30, no. 33, pp. 4696-4698.

Metzger, A., Verheggen, B., Dommen, J. et al. (2010) Evidence for the role of organics in aerosol particle formation under atmospheric conditions, Proc. Nat. Acad. Sci, vol. 107, no. 15, pp. 6646–6651.

Myhre, G., D. Shindell, F.-M. Bréon, et al., (2013) Anthropogenic and Natural Radiative Forcing. In: Climate Change 2013: The Physical Science Basis. Contribution of Working Group I to the Fifth Assessment Report of the Intergovernmental Panel on Climate Change, Cambridge University Press, Cambridge, United Kingdom and New York, NY, USA. https://www.ipcc.ch/site/assets/uploads/2018/02/WG1AR5_Chapter08_FINAL.pdf

Ohmura, A. (2009) Observed decadal variations in surface solar radiation and their causes J. Geophys. Res., vol. 114, D00D05, doi:10.1029/2008JD011290.

Seinfeld, J. H. and Pandis, S. N.: Atmospheric Chemistry and Physics: From Air Pollution to Climate Change, Wiley-Interscience, 1326~pp., 1998.

Volz, F. (1970) Atmospheric turbidity after the Agung eruption of 1963 and size distribution of the volcanic aerosol J. Geophys. Res., vol. 75, no. 27, pp. 5185-5193.

---

## Author Comment (AC1) · 10 Sep 2019

**Response to comments and suggestions from Referee #1**

We thank Referee 1 for insightful comments to and substantial time invested in this MS. We have outlined the answers following the order of comments given by the referee. The three main areas of concern raised by the referee includes how changes in precipitation and cloudiness may affect the result, the fact that we sometimes allowed room for over-emphasizing the results and method descriptions. These main points are treated in individual sections 1-3 below followed by specific comments under section 4.

We believe that the comments by Referee 1 substantially have contributed to a better MS, as well as improved accessibility to the methods employed in this study.

The revised MS is attached to this document, and all revisions are highlighted in red. We also added a supplementary document to better respond to the questions raised by the referees

1.) Changes in clouds and precipitation

**Answer:** We agree that this indeed would be useful information to add to the study. Initially, the idea was to include also statistics with respect to integrated precipitation along trajectories for e.g. 24, 48, 72 and 120h prior arrival. When performing this test, we did observe no obvious change in the in integrated precipitation from this precipitation data.

The first cluster (Cluster 1) was however found to be the cluster most likely resembling ageing under the influence of clouds and precipitation.

What we did not discuss was however whether the enhanced occurrence of size distributions belonging to Cluster 1 was caused by lower emissions alone.

We revisited the data and made an analysis of average precipitation intensity along Cluster 1 (i.e. washout), creating hourly resolved bar-graphs for pre-2009 and post-2008 periods.

In newly added Supplementary material, Figure S1 and Figure S2 together with highlighted text below under sections 3.2 and 4.5 we attempt to address this issue. We also included two supplementary figures connecting to the issue at hand:

"The precipitation intensity along trajectories belonging to each one of the major cluster groups Washout, Nucleation, Intermediate and Polluted, was averaged and the result was a bar graph was used to illustrate the resulting average precipitation during the last 10 days of transport. It can be clearly seen in Supplementary material Figure S1, that the integrated precipitation is largest for Washout and Nucleation type clusters, and smallest for Intermediate and Polluted. This suggest that the Washout-type cluster is indeed more likely to have experienced higher precipitation amount en route to Aspvreten.  The Nucleation-type clusters exhibit an interesting pattern: precipitation rate is on average high up to some 20h prior arrival where after average intensity decreases. We hypothesize that washout followed by a clearing of skies just before arrival paves way for new particle formation, which in turn highlight the need for both relatively clean air (low condensation sink) together with high photochemical activity to create favorable conditions for new particle formation in the size range of a few nanometers. Hence, the epithet names given to the cluster types is

consistent with the expected relation to the evolution of precipitation along the trajectories (Tunved et al. 2004)

"We further attempted to relate the increased relative frequency of Washout-type clusters to trends in precipitation amount along the trajectories. The analysis did not result in any clear time dependent trends. In order further investigate the relative roles of wet removal and source strength, we revisited the data and made an analysis of average precipitation intensity along Cluster 1 (i.e. Washout-type cluster), creating hourly resolved bar-graphs for pre-2009 and post-2008 periods, to see if there exist any differences. If post 2008 data contain trajectories with on average more precipitation, enhanced wet removal may play an important role. It is clear, that integral precipitation on average is about the same for Cluster 1, but slightly different patterns in timing of precipitation can be noted. This tells us that at least for the last ~2 days, we have, if anything less precipitation influence. This in turn do suggest that the relative increase of size distribution observations belonging to Cluster 1 is not likely to be the result of increase in precipitation, and if we assume that the sink strength would be the same, the redistribution of cluster members in favor of Cluster 1 must be dominated by  emission reductions and not enhanced sink processes."

2.) Problems with over-emphasizing the results.

-2.3 reg. clustering assumptions

**Answer:** Certainly, the number of clusters will to somewhat degree impact the results. The idea of starting with a large number of smaller clusters was to identify, with somewhat high level of detail, the major aerosol size distribution types observed under the 17 years of observations. This type of clustering has been performed in several previous studies and have yielded similar results. Tests was performed with both fewer (8) and more clusters (up to twenty), and the optimal balance between data amount and information content was found to be in the order of 12 clusters. We added following text to the material under section 2.3:

"Test were performed with both more and fewer clusters. The best balance between information content and data amount to be presented was found to be reasonable around twelve clusters. A balance of information content refers to be able to follow a logical context between different clusters, but also that the whole set of clusters can characterize the domain of different size distributions observed at Aspvreten.
"

-    P8, 1.32: Other causes for absence of smaller particles.

**Answer:** We agree. Reworded as:

"Absence of small particles suggest reduced amount of nucleating species likely due to reduction in photochemical activity and/or seasonal changes primary sources."

- Figure 5 skewed due to amount of light at different hours and season

This is a good point. However, this has implicitly already been addressed in the original MS. Figure 4 shows the seasonal distribution of the clusters, and as can be seen, all nucleation-type clusters exhibit similar seasonal pattern. If however, the different nucleation type clusters would be season specific, concerns regarding skewing or biasing would be relevant.

- The Cluster history in Section 3.2.

**Answer:** We have increased the analysis to include also results from analysis of precipitation history, which to some extent provide better support to our conclusions. We also added, as suggested, a clarification that much of the discussion is based on general knowledge regarding aerosol tropospheric lifecycle.

"Much of the reasoning above is based on knowledge of the tropospheric aerosol knowledge. Some additional support to the analysis was given by the precipitation history of trajectories belonging to the 4 major cluster groups. The precipitation intensity along trajectories belonging to each one of the major cluster groups Washout, Nucleation, Intermediate and Polluted, was averaged and the result was a bar graph was used to illustrate the resulting average precipitation during the last 10 days of transport. It can be clearly seen in Supplementary material Figure S1, that the integrated precipitation is largest for Washout and Nucleation type clusters, and smallest for Intermediate and Polluted. This suggest that the Washout-type cluster is indeed more likely to have experienced higher precipitation amount en route to Aspvreten. The Nucleation-type clusters exhibit an interesting pattern: precipitation rate is on average high up to some 20h prior arrival where after average intensity decreases. We hypothesize that washout followed by a clearing of skies just before arrival paves way for new particle formation, which in turn highlight the need for both relatively clean air (low condensation sink) together with high photochemical activity to create favourable conditions for new particle formation in the size range of a few nanometres. "

- P12, 1.1. It was not the intention to exclude other processes.

**Answer:** We highlight this by making following rewording:

"The gradient in trend is further very sharp between 40 and 80 nm. The presence of the two modes could be suggestive of mainly two different processes acting independent on the size distribution."

- Figure 14 and relevance of estimating cloud droplet number under the assumption of liquid phase only.

**Answer:** We agree that this could have been discussed more thoroughly. The assumption of liquid phase only most likely overestimates the number of droplets due to the lack of competitive growth in mixed phase clouds. Thus, the estimates presented by us represent an upper limit for the CCN number derived based on variable used to initiate the model e.g. (Lohmann and Feichter, 2005). In clouds that are warmer than -36C, formation of ice crystal occurs on so called ice nuclei , typically mineral dust and soot particles of larger diameters (DeMott et al., 2010), although other particle types may serve as ice nuclei as well (e.g. (Murray et al., 2012). In (Kanitz et al., 2011) Figure 3 presents observed fraction of ice-containing clouds in different environments function of cloud top temperature. As can be seen, there is a high degree of variation in fraction of ice containing clouds depending on environment studied. For e.g. the Leipzig dataset, 70% of the clouds contain ice at -19°C, while similar ice fraction is only achieved at -34°C in the Punta Arenas dataset.

If we assume our calculated estimate is restricted to low level clouds, and given that the average surface temperature at Oxelösund (a few km south-west of Aspvreten) are below zero degrees Celsius during winter months only (data source: Swedish Meteorological and Hydrological Institute, SMHI, www.smhi.se, 1961-2018), the problem arising from mixed phase clouds is likely contained to the winter months December-January. Thus, our assumption regarding liquid phase only cloud droplet activation and growth is likely prone to errors during this season and is likely to overestimate the CCN number in our idealized simulations.

One has however to bear in mind that the presented results are based on highly simplified assumptions regarding both chemistry and dynamics of the clouds (e.g. fixed updraft, no entrainment, liquid phase only). Typically, CCN concentration is often presented as the number concentration above some predefined size range, often N above 100nm. The idea behind presenting the result as calculated CCN concentration was to allow more influence of shape and concentration of the size distribution on the resulting CCN. The approached used in the current study is of course not flawless, and large uncertainty remain. We do however believe that it represents a better way of estimating CCN than simply applying a fixed threshold.

Nature of mixed phase clouds is not well understood but using average temperature as guidance where surface temperature does not get lower than 0 deg C except during DJF. The effect of IN would therefor be largest during this season. Accounting for the effect of mixed phase cloud activation is rather complicated and deserves a dedicated study. Introducing the use of a cloud parcel model in this study to simulate activation and growth in warm clouds presents a better estimate of CCN's compared to an approach using fixed cut-off diameter only (e.g. 100nm). It is now clearly stated that our estimate omits mixed phase cloud dynamics, and thus the calculated CCN may be misleading during the colder months of the year.

The two following paragraphs was included in the revised MS:

"This is a highly generalized assumption that neglects the influence of competitive growth in mixed phase clouds. The introduction of ice crystals into a pure liquid phase cloud causes the ice crystals to grow on expense of the liquid phase due to the relatively lower saturation

vapor pressure over ice compared to liquid droplets. In clouds that are warmer than -36C, formation of ice crystal occurs on so called ice nuclei , typically mineral dust and soot particles of larger diameters (DeMott et al., 2010), although other particle types may serve as ice nuclei as well (e.g. (Murray et al., 2012). In (Kanitz et al., 2011) Figure 3 presents observed fraction of ice-containing clouds in different environments function of cloud top temperature. As can be seen, there is a high degree of variation in fraction of ice containing clouds depending on environment studied. For e.g. the Leipzig dataset, 70% of the clouds contain ice at -19°C, while similar ice fraction is only achieved at -34°C in the Punta Arenas dataset.

If we assume our calculated estimate is restricted to low level clouds, and given that the average surface temperature at Oxelösund (a few km south-west of Aspvreten) are below zero degrees Celsius during winter months only (Data source: Surface temperature, hourly values, from Swedish Meteorological and Hydrological Institute, SMHI, www.smhi.se, 1961-2018), the problem arising from mixed phase clouds is likely contained to the winter months December-January. Thus, the reader should bear in mind that our assumption regarding liquid phase only cloud droplet activation and growth is likely prone to potentially larger errors during this season, resulting in an overestimate of the CCN number in our idealized simulations. "

- Section 4.5, regarding the use of HYSPLIT trajectories.

**Answer:** We agree that individual trajectories are error prone, and this is especially true in regions where data sources for evaluation of the meteorological model is sparse. In this study we have however used a very substantial number of trajectories, over 130000. Although the accuracy of individual trajectories is somewhat questionable, we believe that the large number of trajectories, and the fact that we are studying intra-model variability, the systematic pattern we present as our main finding in Figure 15 and Section 4.5 would likely not be a coincidence. In order to meet with the referee halfway on this issue we add:

"It should also be added, that there is a substantial uncertainty in calculation of trajectories, especially in regions where possibilities of meteorological model validation are poor like in the Arctic region. The results presented here do however indicate that if changes in transport pattern in fact take place, the changes over the studied period seems small, or non-linear."

- P17, 1.9:

**Answer:** We agree. Our statement is not supported by our analysis. We exclude "natural and anthropogenic" from the sentence. It now reads:

"The data has been analyzed from several different aspects in order to shed light on the degree, when and where changes in emissions may have had an impact on the observed aerosol size distribution."

- P 13, 1.28: We have changed the sentence in accordance with the suggestion by the referee.

**Answer:** It now reads:

"Based on this calculation, we estimate that mass was reduced by roughly 52 % during the studied period."

- P. 18, l. 1-8: How would changes in precipitation impact these findings, if at all?

**Answer:** This question relates to how changes in precipitation would change the new particle formation potential in terms of balance between generation and removal (condensation sink, CS) of nucleating species. In principle, a similar result could possibly be the result of enhanced removal of both nucleating species and condensation sink. Increased precipitation would enhance the removal of both accumulation mode (i.e. CS) as well as precursor gases such as $SO_2$. However, according to the statistics of precipitation en route to Aspvreten, precipitation has likely decreased if any change has occurred (c.f. newly added supplementary material Figure S2) and previously outlined answers.

We do however highlight this by adding after lines 1-8, on page 18:

"Another possibility that we cannot rule out is that changes in precipitation during the studied period have resulted in more efficient removal of both accumulation mode particles and nucleating species. Recalling Figure S2 in Supplementary material, any trend as indicated by trajectory derived precipitation includes a decrease or redistribution along the transport paths comparing pre-2009 to post 2008 periods. As we have not explicitly studied cloudiness over the site, we can neither rule out the possibility that on average increased cloudiness have reduced the photolysis rate close to Aspvreten. Both the role of precipitation and cloudiness in changes of new particle formation would likely deserve a dedicated study."

3.) In this part of our response we outline our answers to the questions relating to the occasionally insufficient description of methods.

- Basic information regarding the sampling is missing.

**Answer:** In fact, this basic information is present in the original MS, Section 2.1, first paragraph reads:

"Aspvreten observation station (58.8°N, 17.4°E, 25m asl) is located in the county of Södermanland, about 80 km south of Stockholm. The station is situated close to the Baltic Sea, and the coastline is located a few km to the East and about one km to the south of the station. The surroundings are dominated by deciduous and conifer forests, in mosaic with farmlands. The station represents typical continental rural background conditions with few local pollution sources or densely trafficked roads. Climatologically, the surroundings represent the boreal-nemoral zone, a transition region between the temperate southern nemoral zone (mainly deciduous broadleaf forest) and the boreal zone. "

-Regarding bias of estimated trends resulting from calibration error, what measures has been taken to assure good quality of data.

**Answer:** During the entire period of interest, the instrument has been maintained by the same technician, Hans Karlsson. This by itself assures consistent calibration routines and standard operating procedures. Further, the DMPS-system have been part of the European Supersites for Atmospheric Aerosol Research (EUSAAR, http://www.eusaar.net/) intercalibration workshop and the setup follows the recommendations made by EUSAAR regarding sampling and inversion. Thus, we are quite confident that the database is of good quality, and that this certifies that any error introduced by changes in sampling routines, inversion or other handling is small.

We added under methods:

"During the entire period of interest, the instrument has been maintained by the same technician. This by itself assures consistent calibration routines and standard operating procedures. Further, the DMPS-system have been part of the European Supersites for Atmospheric Aerosol Research (EUSAAR, http://www.eusaar.net/) intercalibration workshop and the setup follows the recommendations made by EUSAAR regarding sampling and inversion. This certifies that any error introduced in the trend analysis by changes in sampling routines, inversion and such is small."

Furthermore, (Glantz et al., 2019), found a significant decrease in Aerosol Optical Thickness (AOT) over the western Gotland basin area of 1.5%, 1.1% and 1.6% per annum derived from MODIS c051, MODIS c061 and AERONET GDT, respectively. This would translate to an overall decrease over the 17 year period covered by our study of about 25%.

- Table 1-2. Define GSD and Dg

We have added in the caption of Table 1 and 2:

"GSD represents the geometric standard deviation and Dg the geometric mean diameter of each one of the log-normal modes."

and on Page 6, section 2.2 we reworded a sentence to read:

"…henceforth denoted GSD, giving an estimate of the width of each log normal mode centered around $\overline{D}_{pg}$."

- Figure 7: Reversed Y-axis and reference for how CI was calculated.

**Answer:** We agree, after revisiting Figure 7, that the reverse order of the left y-axis disturbs the picture. It is now changed to normal. Regarding the reference for the CI interval, we have used the software package ktaub.m for most of the statistical calculation. This can be found, with open access on mathworks.com user community. In the original MS we forgot to reference it, and now we have included this reference in the text and in the list of references. Full reference of statistical package is "Burkley, Jeff, A non-parametric monotonic trend test computing Mann-Kendall Tau, Tau-b, and Sen's Slope written in Mathworks-MATLAB implemented using matrix rotations. King County, Department of Natural Resources and Parks, Science and Technical Services section. Seattle, Washington. USA. http://www.mathworks.com/matlabcentral/fileexchange/authors/23983."
The confidence intervals of the Sens's slope are calculated following Hollander, M. and Wolfe, D. 1973, Chapter 9, pp. 207-208. This is now properly referenced in the revised MS.

- Regarding Figure 8 and associated confidence intervals and significance.

**Answer:** Figure 8 has been reworked slightly to include the areas of the figure showing trends are significant at 95% confidence level. Please note the small changes in the plot. The reason being us changing the underlying data set used to daily average instead of hourly. This change was done in order to minimize the ties in the data. However, this did not change the results. The areas of the new figure enclosed by dashed red line represents data pairs month/size that did not pass test for significance.

**Answer:** Figure changed to include display where trends were found significant at 95% confidence level. In the legend of Figure 8 we added:

"Daily averaged data was used for calculation of Theil-Sens's slope. Areas enclosed by the dashed red line represents pairs of month/size bin where test for significance failed."

Upper and lower confidence bounds are now depicted in supplementary material Figure S3.

- P12, 1.33, Table 3.

**Answer:** We added in the text:

"All trends of number concentrations were found significant when applying seasonal Kendall test following the method proposed by (Hirsch and Slack, 1984). The choice of seasonal Kendal test was motivated by the serial dependencies present in the data set (i.e. seasonal variation)."

- Figure 9: Please explain in the text how the confidence intervals for the slope were calculated. Please indicate in the caption what the whiskers indicate (range of the data? SD? Etc.).

**Answer:** The figure caption now reads:

"Figure 9: Trends of fitted log normal parameters 2000-2017 of nuclei, Aitken and accumulation modal diameter and modal concentration. Frame a-c shows trends for fitted geometric modal diameter and frame d-f shows trends for fitted modal number concentration. Whiskers indicate 25-75$^{th}$ percentile of data. Fitted trend (as Theil-Sen estimator) slope is given in each title. Upper and lower 95% confidence intervals of slopes are indicated in figures. CI of slopes calculated following Hollander and Wolfe (1973)."

- Figure 10: Indicate what the whiskers indicate.

**Answer:** Caption now reads:

"Figure 10: Theil-Sen estimator slopes for annual trend per month for fitted number concentration of nuclei, Aitken and accumulation modes, January-December 2000-2017. Indicated by error bars are upper and lower confidence intervals of slopes at σ(0.05). All trends tested with Mann-Kendall test and shown significant for σ=0.05 except Aitken mode trend during June-July and November. Upper and lower CI calculated following Hollander and Wolfe (1973) "

- Figure 11: Indicate what whiskers show.

**Answer:** Caption of Figure 11 now reads:

"Figure 11: Box-Whisker plot of daily mean mass concentration 10-390nm assuming density of 1 g cm$^{-3}$ for Aspvreten 2000-2017. Theil-Sen estimator slope and associated 95%CI of slope is indicated in figure by solid black lines. Note that outliers are excluded in the figure for improved clarity. Upper and lower bounds of boxes show 25$^{th}$-75$^{th}$ percentile range, and whiskers 5$^{th}$-95$^{th}$ percentile ranges of data. Red lines show median. "

4.) Specific comments

- "Clearly there is a lack of long-term aerosol microphysical data from which trend analysis has been reported or even can be reported because of too short measurement periods or interrupted data sets." To clarify, can the authors please state whether there are similar observations in other locations, and/or what specific region the above statement is referring to?

**Answer:** We reworded this sentence to be more positive.

"Clearly there is a lack of long-term aerosol microphysical data from which trend analysis has been reported or even can be reported because of too short measurement periods or interrupted data sets. Trend analysis require long term uninterrupted time series to be meaningful. This does not mean that data is completely missing. Through collaborations within European Supersites for Atmospheric Aerosol research (www.EUSAAR.net), long term standardized observations of e.g. aerosol number size distribution observations are available from several European sites, some of which are long enough to support time series analysis and determination of trends (e.g. Asmi et al., 2013). "

- Fig. 6: please add the location of the sampling site to this map.

**Answer:** Figure has been updated.

**-** P. 12, l. 2: "As sulfur is the particle precursor that has decreased the most …" please cite what information this statement is based on.

**Answer:** Reference added. Sentence now reads:

"Sulfur emissions in Europe have seen a dramatic decrease during the last decades ((Smith et al., 2011;Aas et al., 2019). Being one of the prime precursors for formation of aerosol number and mass formation, we attribute the strong negative trends around 80 and 200nm to a reduction in aerosol sulfate."

- P. 16, l.2: "However, our analysis above also emphasize that the apparent trends are not the sameover the whole year." Apparent trends in what?

**Answer:** Reworded sentence as:

**"**However, our analysis above also emphasizes that the calculated trends in size distribution properties are not the same over the whole year."

- p. 17, l. 15: "The general trend found in this study is well in agreement with the findings presented by Asmi et al. (2013)…." Please mention here where the Asmi et al. study took place

**Answer:** This was added in the first revision. It reads:

"The general trend found in this study is well in agreement with the findings presented by (Asmi et al., 2013), and extrapolation of their derived trends (N20, Asmi et al., 2013, Table 2) over the same time period presented in this study would give results in the range of 23%, 20% and 40% at Vavihill, Hyytiälä and Värriö and an increase of 2% at Pallas. A simple estimate of mass trends based on the presented trends for N100 yield values similar to our estimates of sub-micron mass trends found in the current study. "

Literature

Aas, W., Mortier, A., Bowersox, V., Cherian, R., Faluvegi, G., Fagerli, H., Hand, J., Klimont, Z., Galy-Lacaux, C., Lehmann, C. M. B., Myhre, C. L., Myhre, G., Olivie, D., Sato, K., Quaas, J., Rao, P. S. P., Schulz, M., Shindell, D., Skeie, R. B., Stein, A., Takemura, T., Tsyro, S., Vet, R., and Xu, X. B.: Global and regional trends of atmospheric sulfur, Scientific Reports, 9, 10.1038/s41598-018-37304-0, 2019.

Asmi, A., Coen, M. C., Ogren, J. A., Andrews, E., Sheridan, P., Jefferson, A., Weingartner, E., Baltensperger, U., Bukowiecki, N., Lihavainen, H., Kivekas, N., Asmi, E., Aalto, P. P., Kulmala, M., Wiedensohler, A., Birmili, W., Hamed, A., O'Dowd, C., Jennings, S. G., Weller, R., Flentje, H., Fjeraa, A. M., Fiebig, M., Myhre, C. L., Hallar, A. G., Swietlicki, E., Kristensson, A., and Laj, P.: Aerosol decadal trends - Part 2: In-situ aerosol particle number concentrations at GAW and ACTRIS stations, Atmospheric Chemistry and Physics, 13, 895-916, 10.5194/acp-13-895-2013, 2013.

DeMott, P. J., Prenni, A. J., Liu, X., Kreidenweis, S. M., Petters, M. D., Twohy, C. H., Richardson, M. S., Eidhammer, T., and Rogers, D. C.: Predicting global atmospheric ice nuclei distributions and their impacts on climate, Proceedings of the National Academy of Sciences of the United States of America, 107, 11217-11222, 10.1073/pnas.0910818107, 2010.

Glantz, P., Freud, E., Johansson, C., Noone, K. J., and Tesche, M.: Trends in MODIS and AERONET derived aerosol optical thickness over Northern Europe, Tellus Series B-Chemical and Physical Meteorology, 71, 10.1080/16000889.2018.1554414, 2019.

Hirsch, R. M., and Slack, J. R.: A NONPARAMETRIC TREND TEST FOR SEASONAL DATA WITH SERIAL DEPENDENCE, Water Resources Research, 20, 727-732, 10.1029/WR020i006p00727, 1984.

Kanitz, T., Seifert, P., Ansmann, A., Engelmann, R., Althausen, D., Casiccia, C., and Rohwer, E. G.: Contrasting the impact of aerosols at northern and southern midlatitudes on heterogeneous ice formation, Geophysical Research Letters, 38, 5, 10.1029/2011gl048532, 2011.

Lohmann, U., and Feichter, J.: Global indirect aerosol effects: a review, Atmospheric Chemistry and Physics, 5, 715-737, 10.5194/acp-5-715-2005, 2005.

[revised manuscript text omitted]

Supplementary material:

[Figure]

Figure S1: Average of 10 days precipitation along trajectories connected to main cluster groups Washout, Nucleation, Intermediate and Polluted. Aspvreten, 2000-20017.

[Figure]

Figure S2: Average hourly precipitation along 10-day trajectories arriving Aspvreten pre-2009 and post-2008. See text for further details.

[Figure]

**Figure S3:As Figure 9, but showing 95% Confidence interval of Theil-sens slopes for lower (left) and upper (right) confidence interval. Color indicate calculated linear trend for binned particle number concentration at Aspvreten as particle cm⁻³ year⁻¹ for the time period 2000-2017.** **Daily averaged data was used for calculation of Theil-Sens's slope. Areas enclosed by the dashed red line represents pairs of month/size bin where test for significance failed.**

---

## Author Comment (AC2) · 10 Sep 2019

**Response to comments and suggestions from Referee #2**

We thank referee 2 for the time invested into this manuscript. We have below outlined our response in detail. Based on the general comments we have tried to isolate the main concerns raised by referee 2 into 6 major replies as outlined below. We cannot agree with every one of these main comments, although we have tried to adjust the manuscript according the suggestions made by the referee as far as possible. This is a rather large manuscript and including even more details (e.g. more extensive discussions and figures relating to new particle formation) is beyond the scope of this paper. We still however believe, that the insightful comments by the referee has helped us to substantially improve the MS.

Regarding the more specific comments, 1-32, we have outlined our response in a similarly structured way so that the editor and referee easily can identify our response.

The revised MS is attached to this document, and all revisions are highlighted in red. We also added a supplementary document to better respond to the questions raised by the referees

**General comments:**

1.) The referee raises concerns regarding the Abstract and Section 1. The reviewer suggests a comparison plot that put the Aspvreten site and NPF events in context to other Scandinavian sites.

**Answer:** This study explores trends in a 17-year long data set. As nucleation is such a prominent feature of the aerosol in the nemoral-boreal region over Scandinavia, discussion regarding nucleation is unavoidable. A climatology and trend analysis of NPF events at Aspvreten is the focus of a separate study currently underway. The focus of the current paper is however not to study nucleation per se.

Regarding comparison with other stations, please c.f. p3 l. 23 & onwards and p4 l1-11. We clearly put the station in context by providing a summary of studies of aerosol properties from a number of different sites across Sweden and Finland. (e.g Kristensson et al. (2008), DalMaso et al. (2005), Laakso et al. (2003), Tunved et al. (2003) and Kulmala et al. (2008) and Komppula et al. (2006). We also compare our findings regarding the trends with Asmi et al. (2013). See p 17 l. 15 onwards, and have included comparison with trends in PM2.5 observations derived by Törseth et al. (2012). Our opinion is that the observations at Aspvreten are duly compared to other Scandinavian sites in the current version of the manuscript, and the reviewer will have to wait for a paper with a specific NPF focus. We therefore cannot agree with the referee that this information is missing.

2.) Regarding modes of figure 2, we understand that the question such that the reviewer expect the modes present in the in Tables also should be visible also in the plots of the average size distribution.

**Answer:** Log normal fitting is a tool used to condense multivariate size distribution information into a more manageable and easily accessible form. Section 2.2 describes this methodology. We clearly state that we use three modes, and perform a unconstrained fit over the size range covered by the instrument, where after we arrange the modes according to their size into mode 1-3 (or nuclei, Aitken and accumulation modes). However, if the *average* of the *size distribution* for any given period is compared to the size distribution calculated from the *average* of *fitted parameters*, one cannot expect to get agreement.

The fact that we might have an average of raw size distributions that appear bi-modal, is not the same thing as saying that the aerosol contains two modes only even though we could reach a conclusion that two modes would suffice. This is however not how we have done it. To highlight this point, we have calculated the average size distribution resulting from the individual fits and superimposed them on the same statistics resulting from averaging of raw size distributions during summer. We have done it for the other seasons as well, but limit figure below to the summer. This compares Figure 2, Summer (second frame) with same statistics derived from the fitted data, i.e. the log-normal parameters for each, in this case hourly, distribution has been used to calculate number size distributions following eq. 1 Section 2.2. Result presented below for summer months as median and 25th-75th percentile ranges:

[Figure]

Figure 1: Comparison between calculated median and 25th-75th% ranges for original size distribution data (blue surface, dashed lines)  and same metrics derived from the individual fits (red surface, solid lines).

Slight deviations are present as the simplification into three modes does not always result in perfect fit.

3.) Continuing from point 2, regarding the diurnal variation during new particle formation events and the usefulness to numerical modellers.

**Asnwer:** We agree with the statement of the referee.

4.) Regarding frequency of nucleation events at Aspvreten.

**Answer:** This is a good point that deserves more analysis as well, but we feel it is not within the scope of this paper. We are planning to make a more detailed study of new particle formation in the future to map nucleation event characteristics for the 17 year of data collected. Regarding the use of the data within the modeling community, the underlying data set has been published at the PANGAEA data repository under open access. Common statistical data for size distributions observed at different stations see response to comment 1. For modelers to compare with our result in this study, they need to treat their results in an analogous procedure.

5.) Regarding the calculations of cloud droplet activation using the adiabatic cloud parcel model in CALM (Tunved et al. (2010)).

**Answer:** We have used the adiabatic cloud parcel model to estimate the potential activation of CCN's under different assumptions regarding chemistry and updraft. This was done in order to try to estimate to what extent the observed trends in aerosol number size distribution properties translate into potential effect on clouds. Section 2.3 of Tunved et al. (2010) provides a description of the adiabatic parcel module for calculation of activation and droplet growth (in that particular case it was used to estimate the role of liquid phase oxidation of sulfur dioxide).

Although our approach is limited to liquid phase clouds only, we still think that the calculations presented in the current version of the MS do add value to the overall findings. Some improvements in the text have been added as per request of referee 1, which have moderated our previous statements in order to avoid overemphasizing of our results.

We are aware that an approach assuming liquid phase only, no entrainment and assuming constant updraft within cloud etcetera could provide somewhat schematic results. With this said, we still however believe that our approach provides more information than simply integrating the number concentration above some certain size limit, e.g. CCN~N(Dp>100nm).

Thus, with edits following the suggestions by Referee 1, we suggest to keep this section even though estimates contain uncertainties.

6. The reviewer suggest to restrict the study to aerosol trends only.

To assess the sensitivity and potential importance of observed NSD trends on radiative budget we performed abovementioned simulations. Note we did not beforehand know if the results would come out the way they did. If we would have reach the conclusion that the changes haven't impacted cloud formation, it would of equal importance for the study. Thus, we disagree with the referee.

In summary, a dedicated study focusing on NPF at Aspvreten during the period 2000-2017 is underway. However, to include the rigorous analysis required to perform such a study to be included in the current MS would make an already extensive study manuscript too long.

**Response to specific comments**

1.) Investigate→investigated
2.) We have changed sentence to read: "We show that both particle modal number concentration and size substantially has been reduced during last 17 years.". The fact that changes occur in both number and size have already been mentioned.
3.) We moved the sentence "These decreases are similar to observations found at other stations in Northern Europe." to the end of this paragraph in order to improve readability, and rewrote from singulars to plurals wrt to decreases.
4.) This is already covered in the abstract although not in the suggested order. We see no action required.
5.) Covered in sentence following l. 19-22. C.f. l. 23-24.
6.) Sentence reworded. No reads: "The combined trajectory and data analyses do not present evidence for an increase in new particle formation formed locally,…"
7.) Yes. See answer 5, major comments (CALM-model)
8.) We disagree as outlined in major comments above.
9.) We also thought that this was a good alternative word to start the introduction.
10.) Now changed to comply with suggestion
11.) Global (or perhaps atmospheric) diming would be a more appropriate word. In this context solar output is more or less constant, so we would like to avoid to "solar dimming".
12.) We rewrite sentence as: "…Earth's surface indicate that global dimming increased up to about 1990…". Also rewrote following sentence as: "…Streets et al. (2006), suggested that this trend in global dimming/brightening was due to the combined effect of economic growth and the recent decrease in emission of aerosol particles and their gaseous precursors as a result of legislative measures."
13.) "Vapour" was added as per request. The rest of the sentence reads, in our opinion, well as is, i.e. "…participates both in new particle formation and contributes to aerosol mass concentration by condensation on already existing particles."
14.) We feel that the suggestions by the referee regarding this section of the introduction is more a question about taste. It is our opinion that the different areas of the globe are covered adequately in order to put our data into context. This rely also covers following points 16-18
15.) This typo has been corrected.

16.) It is unclear to us what is wrong with the balance of the current sentence: "Coen et al. (2013), studied trends in observed aerosol optical properties measured at ground-based stations mainly located in North America and Europe. They concluded that, even if the trends are not homogeneously distributed geographically, the 25 decreasing trend observed for most stations in North America is related to the decrease in anthropogenic emissions of particles and their gaseous precursors. The pattern of trends over Europe was not as clear as over North America and the Arctic and Antarctic stations did not show significant trends at all"

17.) C.f. 14.

18.) C.f. 14.

19.) Rephrased as "However, due to claimed lack of stations with more than 10 years of data, only five stations were available for their study.".

20.) We have added as per request by referee 1: "Clearly there is a lack of long-term aerosol microphysical data from which trend analysis has been reported or even can be reported because of too short measurement periods or interrupted data sets. Trend analysis require long term uninterrupted time series to be meaningful. This does not mean that data is completely missing. Through collaborations within European Supersites for Atmospheric Aerosol research (www.EUSAAR.net), long term standardized observations of e.g. aerosol number size distribution observations are available from several European sites, some of which are long enough to support time series analysis and determination of trends (e.g. Asmi et al., 2013)."

21.) We believe that the current wording will suffice, p3, l. 26-30: "In this region, several long term monitoring stations exist covering vegetation zones from nemoral to polar sites north of the tree-line,e.g. the southern Sweden nemoral site Vavihill (56.01°N, 13.09°, 172m asl)(Tunved et al., 2003;Kristensson et al., 2008), the southern boreal site Hyyiälä (61.85°N, 24.29°, 179m asl) (Kulmala et al., 2008;Dal Maso et al., 2005)and the northerly located stationsVärriö (67.76°N, 29.61°, 390m asl) and Pallas (67.97°N, 24.12°, 560m asl) (Laakso et al., 2003;Dal Maso et al., 2007;Tunved et al., 2003)."

22.) We added "( e.g. Kulmala et al., 2001 and Kerminen and Kulmala, 2002)."

23.) We agree that we can be more clearer here. We did re-write as "Based on the literature referenced above, several locations exhibit a decrease of aerosol mass observed over populated areas in the last decades and this decrease is related to the decrease in emissions of anthropogenic primary particles and precursor gases."

24.) Agreed, papers now in chronological order.

25.) This was commented also by referee 1, and the final sentence after comments by referee 2 is: "Absence of small particles suggest reduced amount of nucleating species likely due to reduction in photochemical activity and/or seasonal changes primary sources. Sulfuric acid vapour has a short residence time and is rapidly consumed by either gas-particle-formation or deposition."

26.) It is unclear why the current wording wouldn't suffice. This is what we see, a decrease in polluted clusters, together concurrent with an increase of similar magnitude of washout cluster. The trends for other clusters are comparably small. This means most likely that we observe more number size distributions in cluster one, on expense of decreasing number of observations in clusters 10-12. Regarding including actual numbers, we agree with the

referee. Sentence now reads: "At the same time, the most polluted cluster 12 has been reduced from around 5% of observations to around 1% during the period 2000-2017."

27.) In the original MS we write in Section 5. "This increase may reflect improved conditions from a new particle formation perspective, and this may reflect favorable relation between $H_2SO_4$ and CS or some other nucleating species. As this increasing trend is absent for the smallest size classes, we conclude that if this increase is the result of enhanced nucleation, we argue that it must take place upstream of the station rather than in the immediate vicinity in order to reach the size range where we observe the positive trend. In essence; nucleation occurs earlier during the marine-continental transition and this in turn quench further nucleation closer to the station." To guide the reader where to find this, we write in the bullet: "Although we do not find evidence for increased nucleation close to the measurement site, we do find a positive trend for Aitken mode sized particles during summer months

28.) Since the accumulation mode concentration is reducing, we interpret the increase in Aitken particle concentrations as a response to the decrease in condensation sink, leading to more frequent nucleation upstream of the station (c.f. Section 5.)."

29.) We disagree with the referee. It is of central interest to understand how observed changes in both number and size of the aerosol population feed back on the cloud properties. Although our method is somewhat simplified, we still believe that the calculations of CCN can be of interest as an order of magnitude estimate on effects cleaner air has had on the cloud optical properties.

30.) The plot is created by calculating daily averages of all data present. Then, for each day of a standard year, an average day is calculated. This means that ideally, each data point (i.e. size distribution) is calculated as an average for 17 days (one per year). Nucleation events does not occur during same time (i.e. day of year) every year. This means, that when averaging over days, the nucleation features of NSD's will be masked. On should recall that nucleation has a signature in the sub 20nm size range for a couple of hours, then disappears. We also think that the clustering presented in Section 3.2 et seq clearly demonstrates the fraction of observations showing signs of recent nucleation. We can agree that if Figure 1 was the only figure showing size distribution properties, it would be timely to add additional statistics, but in this manuscript, this would only result in additional information, which further is somewhat redundant. Thus, we decide to leave this comment without additional action. C.f. response to major comment number 2 where we explain why modes may be masked in averages. We think point has been proven by showing figure above.

31.) We have included in figure 2 subplots showing dN/dlogDp in logarithmic y-scale. Figure caption changed accordingly. Caption now reads:

"Figure 2: Seasonally average size distributions observed at Aspvreten 2000-2017. Shaded blue area gives 25th-75th percentile ranges, and dashed line median size distribution. Small graphs give size distribution mean diameter (nm) and average total concentration (cm$^{-3}$). Shown in the sub-frames are same data, but now in log-log scale. Spring=March-May; Summer=June-August; Autumn=September-November; Winter=December-February."

32.) The reason that two modes are not obvious in Figure 1& 2 are caused by on the one hand how the fitting is performed (c.f. section 2.2) and on the other hand the overlap regions between the two modes create an impression of a unimodal distribution in this size range.

This can be easily tested by applying Eq. 1 to the data presented in Table 1, which we have done for the spring data in Figure below. As can be seen, the overlap creates an apparent bimodal distribution, although 3 modes are present. Alternatively, we could have applied a more "intelligent" fitting routine, which apart from performing fitting also attempts to minimize the number of modes. This is however not the case in the current study. In conclusion: the mathematical representation of the aerosol size distribution does not imply that what is perceived from an ocular analysis of the same distribution (i.e. apparent presence of two modes only) automatically will be reflected by the algorithm.

[revised manuscript text omitted]

Supplementary material:

[Figure]

Figure S1: Average of 10 days precipitation along trajectories connected to main cluster groups Washout, Nucleation, Intermediate and Polluted. Aspvreten, 2000-20017.

[Figure]

Figure S2: Average hourly precipitation along 10-day trajectories arriving Aspvreten pre-2009 and post-2008. See text for further details.

[Figure]

**Figure S3:As Figure 9, but showing 95% Confidence interval of Theil-sens slopes for lower (left) and upper (right) confidence interval. Color indicate calculated linear trend for binned particle number concentration at Aspvreten as particle cm[-3] year[-1] for the time period 2000-2017. Daily averaged data was used for calculation of Theil-Sens's slope. Areas enclosed by the dashed red line represents pairs of month/size bin where test for significance failed.**

---

## Author Response (AR2)

Response to Referee #2

The authors thank Referee #2 for time invested in the manuscript. We are happy to hear that Referee #2 accept our response in most parts. In essence, the only remaining larger issue relates to clarification of the effect of seasonal averaging of aerosol number size distributions. In the revised manuscript we have added in Supplementary material Figure S1, a diurnal seasonally averaged size distribution that in addition to the clustering highlight the often observed, intermittent new particle formation events present in the dataset.

We added under 3.1 after first paragraph: "It should be noted that seasonal averages of daily mean aerosol number size distributions do not preserve the signature of new particle formation events (for details regarding new particle formation c.f. e.g. Kulmala et al. 2004). The lack of a distinct nuclei mode in Figure 2 does however not imply that nucleation is absent in the data set, but rather suggest that the intermittent behavior and short lifetime of the nuclei mode under conditions characteristic for the Aspvreten station leads to an masking of these features when performing long term averaging. Thus, in Supplementary material, Figure S1, we show Time-of-Day seasonal mean size distributions. As evident, the signature of new particle formation events is present for all seasons except wintertime. 10nm particles are typically observed around noon but grows rapidly into larger size classes during a couple of hours."

[Figure]

Figure S1: Diurnal variation of seasonally averaged mean number size distributions.

As acknowledged by the Referee, long term averaging over daily means tend to mask the presence of particles in the nuclei mode size range. With aforementioned revision, we hope we have highlighted what might seem as a contradiction depending on how data is presented.

We further expanded the discussion about Figure S3 (Now Fig S4). We have added under Section 4.1, last paragraph: "In figure S4, 95% Confidence interval of Theil-Sens slopes for lower (left) and upper (right) confidence interval. Color indicate calculated linear trend for binned particle number concentration at Aspvreten as particle $cm^{-3}$ $year^{-1}$ for the time period 2000-2017. Areas bounded by the dashed red line represents pairs of month/size bin where test for significance was below the 95% threshold."

Below we provide a point-by-point response to the specific comments raised by the Referee

1.) Reference added in revised MS using citation recommended by the reviewer
http://www.climatechange2013.org/images/report/WG1AR5_Chapter08_FINAL.pdf
"Myhre, G., D. Shindell, F.-M. Bréon, W. Collins, J. Fuglestvedt, J. Huang, D. Koch, J.-F. Lamarque, D. Lee, B. Mendoza, T. Nakajima, A. Robock, G. Stephens, T. Takemura and H. Zhang, 2013: Anthropogenic and Natural Radiative Forcing. In: Climate Change 2013: The Physical Science Basis. Contribution of Working Group I to the Fifth Assessment Report of the Intergovernmental Panel on Climate Change [Stocker, T.F., D. Qin, G.-K. Plattner, M. Tignor, S.K. Allen, J. Boschung, A. Nauels, Y. Xia, V. Bex and P.M. Midgley (eds.)]. Cambridge University Press, Cambridge, United Kingdom and New York, NY, USA"

2.) Changed according to referee suggestion.

3.) Now reads: "Seasonal variation of the aerosol number size distribution between 10-390nm presented as daily median aerosol number size distribution for the whole study period, 2000-2017. Superimposed on the surface plot is the median and quartile ranges of integral number concentration."

4.) Figure 2 is adjusted according to the suggestion by the referee.

5.) Caption now revised and reads: "Figure 2: Seasonally average size distributions observed at Aspvreten 2000-2017. Shaded blue area gives 25th-75th percentile ranges, and dashed line median size distribution. Shown in the sub-frames are same data, but now in log-log scale. Spring=March-May; Summer=June-August; Autumn=September-November; Winter=December-February."

6.) Caption now reads "Table 1: Statistics of modal fits per season. Table shows statistics derived from fitted hourly number size distributions. Indicated in table are Nuclei, Aitken and accumulation mode parameters as median and 25th-75th percentile. GSD represents the geometric standard deviation and Dg the geometric mean diameter of each one of the log-normal modes."

7.) We added under section 2.3 "The clustering was performed on hourly averaged data, using options "max iterations" of 10000 and "number of replicates" was set to 10 in Matlab." As suggested by the reviewer we also amended first paragraph of 3.2: "This section describes the results from the cluster analysis. As stated under Section 2.3, the clustering was performed on hourly means, roughly 130000 size distributions. This approach captures signature size distributions in different stages of the aerosol life-cycle, including aerosol number size distribution types that originate from new particle formation events (e.g. Kulmala et al. 2004)."
We also point out that in the original MS the second paragraph, section 2.3, reads: "Contrary to standard averaging of number size distributions, cluster analysis and associated centroids can conserve the shape of the aerosol size distribution. Hence, size distribution clusters represent "signature distributions" that reflect contribution from members that are likely to

have undergone similar processing in the atmosphere prior to observations. Thus, clustering size distribution and combining the cluster analysis with auxiliary parameters, such as trajectory derived source areas, temporal distribution of members and parameters related to sink processes (e.g. precipitation) can provide a deeper insight into the multitude of factors defining the aerosol over time. "

8.) *Please add a sentence near the start of this section re-iterating this point about the additional value/virtue within the cluster analysis technique, that (even if this is not yet explored in this particular manuscript) the approach potentially enables to identify changes in nucleation within the long-term measurement record that exists at Aspreveten and other sites.*

In Section 2.3 and in revised section 3.2 we address this, but we do however agree that this could be re-iterated. We thus add in Discussion: "In addition, cluster analysis of hourly number size distributions has been demonstrated to be a useful tool in trend studies. The method has been applied to study how the aerosol observations are distributed over 12 signature distributions, and further applied to investigate how the aerosol have been re-distributed between these 12 dominating cluster types during the period 2000-2017. The method seems well suited for studying trends in new particle formation events."

9.) First bullet in conclusion reads: "As revealed by the aerosol size distribution clustering, it is evident that the cluster representing clean, cloud processed aerosol is increasing on expense of the polluted type monomodal size distribution. There is only marginal increasing trend of cluster members belonging to clusters showing sign of recent new particle formation. At the same time, the most polluted cluster 12 has been reduced from around 5% of observations to around 1% during the period 2000-2017."

We add to this bullet

"We have shown that cluster analysis successfully can be used to study aerosol trends, and further that cluster analysis also can be used for studying trends of intermittent processes such as new particle formation events. The method has clear advantages compared to standard time averaging techniques as it preserves the shape and number concentration of the aerosol number size distributions within the clusters which otherwise easily can get lost in usually applied time averaging such as mean and medians. "